# mRNA delivery enabled by metal–organic nanoparticles

Yuang Gu [1,3], Jingqu Chen [1,3], Zhaoran Wang [1], Chang Liu [1], Tianzheng Wang [1], Chan-Jin Kim[1], Helena Durikova [1], Soraia Fernandes[1], Darryl N. Johnson [2], Robert De Rose [1], Christina Cortez-Jugo[1] & Frank Caruso [1] ✉

mRNA therapeutics are set to revolutionize disease prevention and treatment, inspiring the development of platforms for safe and effective mRNA delivery. However, current mRNA delivery platforms face some challenges, including limited organ tropism for nonvaccine applications and inflammation induced by cationic nanoparticle components. Herein, we address these challenges through a versatile, noncationic nanoparticle platform whereby mRNA is assembled into a poly(ethylene glycol)-polyphenol network stabilized by metal ions. Screening a range of components and relative compositional ratios affords a library of stable, noncationic, and highly biocompatible metal–organic nanoparticles with robust mRNA transfection in vitro and in mice. Intravenous administration of the lead mRNA-containing metal–organic nanoparticles enables predominant protein expression and gene editing in the brain, liver, and kidney, while organ tropism is tuned by varying nanoparticle composition. This study opens an avenue for realizing metal–organic nanoparticle-enabled mRNA delivery, offering a modular approach to assembling mRNA therapeutics for health applications.

The success of COVID-19 vaccines from Pfizer and Moderna highlights the potential of mRNA-based therapeutics to revolutionize healthcare[1]. Besides vaccines, mRNA agents are poised to make a significant impact in gene editing strategies[2,3] protein replacement therapy[4–6], and personalized medicine[7] to address diverse health challenges, including genetic diseases and cancer. However, achieving efficient mRNA delivery is an ongoing challenge because of the susceptibility of mRNA to enzymatic degradation[8] and limited cellular membrane permeability[9], warranting the need for precisely designed carriers to address these delivery challenges. Current carrier design for mRNA delivery is based on electrostatic interactions between the highly negatively charged mRNA and cationic components (e.g., ionizable lipids[10], polyethyleneimine, and histidine/arginine-rich peptides[11,12]) of nanomaterials that can accommodate a high mRNA payload within a formulated nanoparticle (NP), while also facilitating membrane penetration and endosomal escape[13,14]. However, studies have shown that

the cationic components of these NPs can pose safety concerns by provoking acute neutrophil infiltration and initiating a cascade of inflammatory responses[15–18]. Moreover, NPs predominantly localize to the liver and spleen upon intravenous (IV) administration[19–21], partly due to size-dependent extravasation processes. Once localized in these organs, cationic NPs can exhibit enhanced uptake by phagocytic macrophages and monocytes, which are involved in the clearance of the NPs[20,22–25]. Hence, the development of an mRNA delivery platform devoid of cationic components could potentially reduce premature clearance, facilitate broader organ targeting beyond the liver and spleen, and improve the safety of mRNA delivery.

Polyphenols, a class of naturally abundant molecules, are distinguished by their aromatic carbon structures with multiple hydroxyl groups. This distinctive configuration confers polyphenols with a variety of interactive forces and a high binding affinity toward diverse molecules[26,27]. Moreover, the coordination of polyphenols with metal

[1]Department of Chemical Engineering, The University of Melbourne, Parkville, VIC, Australia. [2]Materials Characterisation and Fabrication Platform, The University of Melbourne, Parkville, VIC, Australia. [3]These authors contributed equally: Yuang Gu, Jingqu Chen. ✉e-mail: fcaruso@unimelb.edu.au

ions under physiological conditions enables the formation of stable, negatively charged metal–organic hybrids termed metal–phenolic networks (MPNs)[28].

In the present study, a combinatorial design is applied to engineer a noncationic and versatile NP platform (i.e., mRNA-MPN NPs) that enables efficient mRNA transfection with tunable organ tropism. The design principle is based on three fundamental elements: (1) the multidentate properties of polyphenols that allow robust mRNA incorporation; (2) metal–phenolic coordination that affords NPs with a rich choice of building blocks and chemical diversity; and (3) a seeding agent that can increase the local concentrations of precursors (mRNA and polyphenols) and drive the formation of NPs under ambient conditions. The engineered mRNA-MPN NPs stabilize mRNA within a metal–organic network and display superior transfection efficiency of mRNA over the commercial transfection agent Lipofectamine in different cell lines. Following IV administration, these NPs enable mRNA delivery in multiple mouse organs, including the liver, kidney, lung, heart, and spleen. Furthermore, protein expression and gene editing from delivered mRNA are observed in the brain. The levels of protein expression are comparable to those achieved by SM-102-containing lipid nanoparticles (LNPs), which are considered the benchmark in nucleic acid delivery. Organ deposition and protein expression occur concomitantly with negligible metal accumulation, tissue inflammation, or cytokine release within the bloodstream. Notably, the organ tropism of protein expression (i.e., organs that exhibit predominant protein expression) can be adjusted simply by altering the composition and ratio of the NP building blocks. The noncationic nature, high biocompatibility, mRNA delivery efficacy, and modularity of the mRNA-MPN NPs not only provide a promising alternative to current mRNA delivery platforms but also open avenues for the development of future NP-enabled therapeutics.

## Results and discussion
### Assembly of mRNA-MPN NPs
MPNs are known for their high modularity, allowing the formation of networks with a combination of different phenolic ligands and metal ions across a range of ratios[29–32]. Given that each formulation exhibits unique noncovalent binding strengths with other molecules[31,32], the first key focus was to determine their compatibility with mRNA. In this context, a systematic screening of combinations comprising poly(ethylene glycol) (PEG, the seeding agent), phenolic ligands, and metal ions was conducted to optimize mRNA incorporation within MPN NPs (Fig. 1a). Briefly, this process entails the sequential mixing of PEG, mRNA, and phenolic ligand to form mRNA–phenolic agglomerates. Subsequent incorporation of metal ions solidifies these aggregates into mRNA-MPN NPs within a time frame of 30 min (Supplementary Fig. 1a).

Epigallocatechin gallate (EGCG, a small phenolic ligand) and $Zr^{IV}$ were first selected as metal–phenolic building blocks for screening the influence of the phenolic ligand-to-mRNA mass ratio on mRNA loading, as this combination enables efficient delivery of proteins and functional small molecules[29,33]. Using a fluorescent protein (mCherry)-encoding mRNA as a model and PEG (20 kDa, linear; referred to as 20k linear) as a seeding agent for mRNA-MPN NP synthesis, the loading of mRNA within the NPs increased from ~20% to ~90% as the EGCG-to-mRNA mass ratio increased from 10:1 to 100:1 (Fig. 1b). The increase in the loading amount is evidenced by the decrease in intensity of the unbound mRNA-related band as shown by agarose gel electrophoresis (Fig. 1b, inset). Accordingly, a phenolic ligand-to-mRNA mass ratio of 100:1 was chosen in subsequent studies. We then studied the effects of different phenolic ligands (tannic acid (TA), EGCG, catechin (CAT), and gallic acid (GA)) on in vitro mRNA transfection. The number of catechol and galloyl groups available for complexation and coordination within the phenolic ligands varies in the order of TA > EGCG > CAT > GA (Fig. 1a). Among the phenolic ligands examined, EGCG resulted in

the highest transfection of HEK 293T cells (i.e., ~95%, which corresponds to the percentage of cells that have been successfully transfected and thus show red mCherry fluorescence), as deduced by flow cytometry (Fig. 1c) and confocal laser scanning microscopy (CLSM) (Supplementary Fig. 2). This is likely attributed to the high loading capacity (Fig. 1d, inset and Supplementary Fig. 3a, b). In contrast, NPs assembled from TA and GA yielded ~40% transfection efficiency, however, with significantly reduced mCherry mean fluorescence intensity (MFI) (Fig. 1d). The mCherry MFI is a measure of the intensity of mCherry expression in each cell (e.g., a low MFI corresponds to a low level of mCherry expression inside the cells). NPs assembled from CAT showed negligible transfection (0.6%), likely owing to both suboptimal mRNA loading (~25%) (Fig. 1d, inset and Supplementary Fig. 3a, b) and compromised particle formation, as suggested by low light scattering in the particle suspension (Supplementary Fig. 4). An in vitro screening heat map showing the influence of the phenolic ligand on mCherry expression is shown in Supplementary Fig. 5, which further demonstrates that higher mRNA transfection efficiencies are obtained when EGCG is used as the phenolic ligand. Regarding the lower transfection efficiency obtained using TA compared to EGCG, the abundant catechol/galloyl moieties of TA might enhance the mRNA loading, but the high binding affinity of TA with mRNA might also impede the intracellular release of mRNA from the NPs[34].

PEG was employed as a seeding agent for mRNA-MPN NP assembly to increase the local concentration of precursors[29], yielding $4 \times 10^9$ mRNA-MPN NPs per milliliter (Supplementary Fig. 1b). Composition screening showed that a high concentration of PEG (i.e., PEG-to-mRNA mass ratio of 100:1) was required for optimal transfection efficacy (Supplementary Fig. 5). Furthermore, the molecular weight ($M_w$) and geometry of PEG may influence mRNA incorporation and transfection efficacy. As shown in Fig. 1e, all PEG variations studied (2k linear, 20k linear, and 2k 4-arm) displayed >50% transfection efficiency. Notably, NP formulations containing either 20k linear PEG or 2k 4-arm PEG resulted in greater mCherry expression, corresponding to cells with nearly a 15-fold higher mCherry MFI than those assembled with 2k linear PEG (Fig. 1f and Supplementary Fig. 6). Longer PEG chains (i.e., higher $M_w$) and a higher number of arms could improve the cross-linking density of the resultant NPs[35,36], which might facilitate the integration of mRNA (Fig. 1f, inset and Supplementary Fig. 3c, d), resulting in increased mCherry expression per cell. As 20k linear PEG displayed the highest transfection efficiency and MFI, it was used along with EGCG in subsequent studies.

We next investigated the effect of the metal ion-to-phenolic ligand (EGCG) mass ratio on mRNA transfection efficacy by assembling mRNA-MPN NPs with different amounts of metal ions while keeping a constant mCherry-to-20k linear PEG mass ratio of 1:100. Compared to the NPs with no metal ion, the inclusion of $Zr^{IV}$ with $Zr^{IV}$-to-EGCG mass ratios up to 1:40 gradually increased the transfection efficiency from 78% to 95% and the MFI by 2.5-fold (Fig. 1g, h and Supplementary Fig. 7a–c). The integration of metal ions also increased the mRNA loading (Fig. 1h, inset and Supplementary Fig. 3e, f). However, increasing the concentration of metal ions beyond a mass ratio of 1:40 inversely impacted the transfection efficiency (reducing to 4% at a $Zr^{IV}$-to-EGCG mass ratio of 10:40) despite the higher mRNA loading observed at higher $Zr^{IV}$ concentrations (Fig. 1h, inset and Supplementary Fig. 3e, f). To investigate this further, mRNA release corresponding to different $Zr^{IV}$-to-EGCG mass ratios was evaluated using fluorescently labeled mRNA. As shown in Supplementary Fig. 7b–d, mRNA release decreased at $Zr^{IV}$ concentrations beyond mass ratios of 1:40 (i.e., ~0.75 μg mRNA was released at 1:40, whereas only ~0.55 μg and ~0.3 μg mRNA was released at 2:40 and 10:40, respectively). This decrease is consistent with the reduction observed in vitro mRNA transfection at the same ratios. This suggests that higher amounts of $Zr^{IV}$ within mRNA-MPN NPs (i.e., $Zr^{IV}$-to-EGCG mass ratio > 1:40) may increase the cross-linking density of mRNA with NPs, thereby hindering mRNA

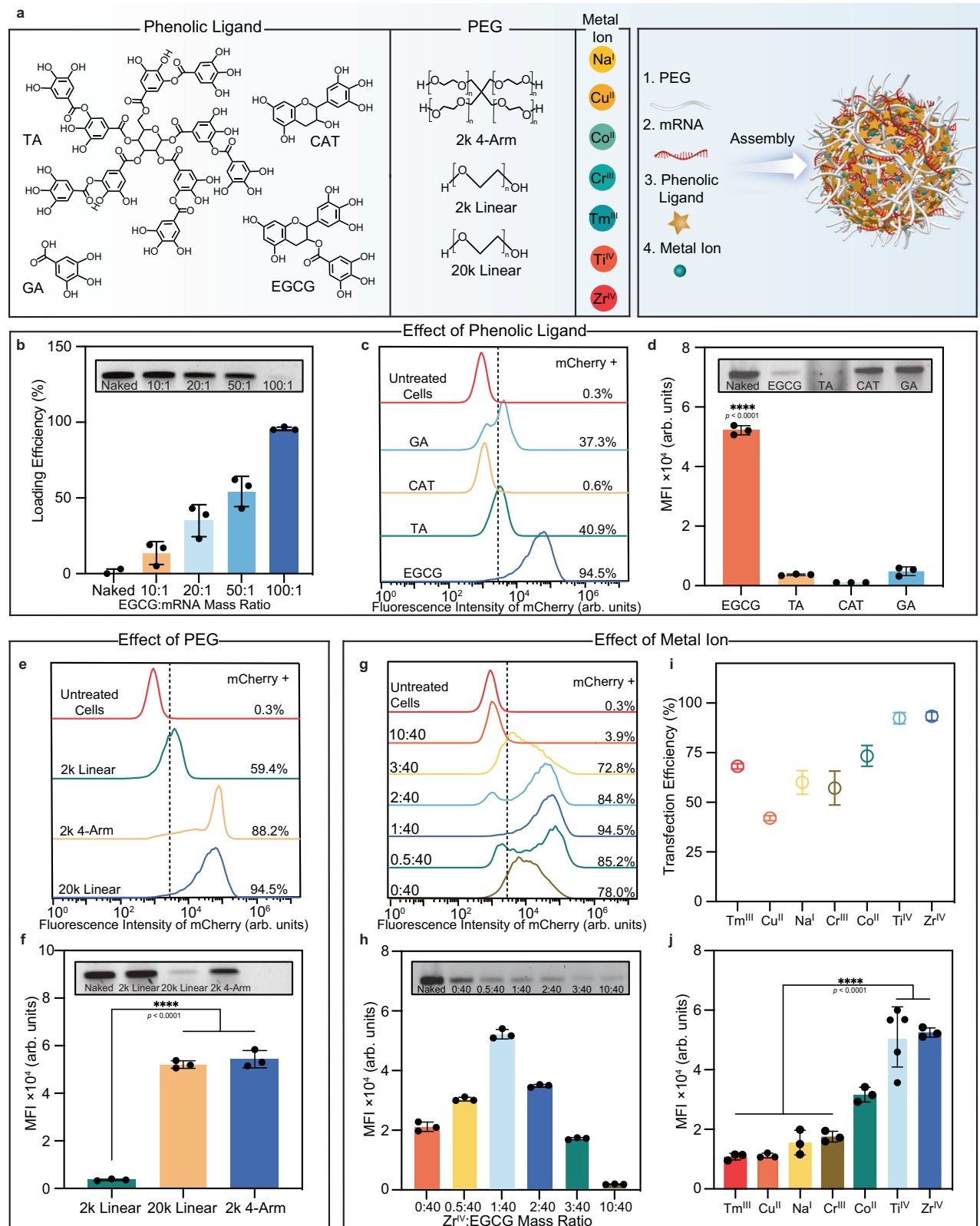

release and reducing the efficacy of mRNA transfection. In addition, a higher metal concentration is likely to induce fluorescence quenching (Supplementary Fig. 8), thus further reducing the MFI of transfected cells. Collectively, the $Zr^{IV}$-to-EGCG mass ratio of 1:40 might exert balanced influences on mRNA loading, release, and fluorescence quenching, making it optimal for mRNA transfection. Other metal ions with different valences and coordination states (i.e., $Na^I$, $Cu^{II}$, $Co^{II}$, $Cr^{III}$,

$Tm^{III}$, and $Ti^{IV}$) were examined, further underlining the versatility of the designed NP platform. As shown in Fig. 1i, j and Supplementary Fig. 9, among the NPs assembled from the range of metal ions studied, those assembled with $Zr^{IV}$ and $Ti^{IV}$ displayed superior transfection efficiency and mCherry expression (i.e., higher MFI).

The NP formulation comprising PEG (20k linear), mRNA, EGCG, and $Zr^{IV}$ at a mass ratio of 100:1:100:2.5 was selected as the lead

**Fig. 1 | Formulation screening and optimization of mRNA-MPN NPs. a** Schematic depicting the synthesis of mRNA-MPN NPs through metal–phenolic-mediated assembly of PEG, mRNA, phenolic ligands, and metal ions. The number indicates the sequence of reagent addition. **b** Loading of mCherry-encoding mRNA in mRNA-MPN NPs (assembled with 20k linear PEG, EGCG, and Zr$^{IV}$) at different EGCG-to-mRNA mass ratios. **c, d** Histogram of mCherry fluorescence (**c**) and mCherry MFI (**d**) of HEK 293T cells transfected for 24 h by mRNA-MPN NPs assembled with different phenolic ligands. $p$ (EGCG vs TA) = $2.3 \times 10^{-11}$, $p$ (EGCG vs CAT) = $8.9 \times 10^{-12}$, $p$ (EGCG vs GA) = $3.5 \times 10^{-11}$. **e, f** Histogram of mCherry fluorescence (**e**) and mCherry MFI (**f**) of HEK 293T cells transfected for 24 h by mRNA-MPN NPs assembled with PEG of different $M_w$ or structure. $p$ (2k Linear vs 20k Linear) = $6.4 \times 10^{-7}$, $p$ (2k Linear vs 2k 4-Arm) = $4.9 \times 10^{-7}$. **g, h** Histogram of mCherry fluorescence (**g**) and mCherry MFI (**h**) of HEK 293T cells transfected for 24 h by mRNA-MPN NPs assembled with different Zr$^{IV}$-to-EGCG mass ratios. The insets in (**b, d, f, h**) show agarose gels with bands representing unbound mRNA (with naked mCherry mRNA, 1056 nucleotides, as a reference, in the first lane of each gel) in the NP formulations studied. **i, j** Transfection efficiency (**i**) and mCherry MFI (**j**) of HEK 293T cells transfected for 24 h by mRNA-MPN NPs assembled with various metal ions; $p$ (Zr$^{IV}$ vs Tm$^{III}$) = $1.1 \times 10^{-6}$, $p$ (Zr$^{IV}$ vs Cu$^{II}$) = $1.2 \times 10^{-6}$, $p$ (Zr$^{IV}$ vs Na$^{I}$) = $5.4 \times 10^{-6}$, $p$ (Zr$^{IV}$ vs Cr$^{III}$) = $1.1 \times 10^{-5}$, $p$ (Ti$^{IV}$ vs Tm$^{III}$) = $3.9 \times 10^{-7}$, $p$ (Ti$^{IV}$ vs Cu$^{II}$) = $4.4 \times 10^{-7}$, $p$ (Ti$^{IV}$ vs Na$^{I}$) = $2.1 \times 10^{-6}$, $p$ (Ti$^{IV}$ vs Cr$^{III}$) = $4.6 \times 10^{-6}$. Data are presented as mean values ± standard deviation (SD), $n$ = 3 or 5. Statistical significance was analyzed using one-way analysis of variance (ANOVA) with Tukey's multiple comparisons test. TA tannic acid, EGCG epigallocatechin gallate, CAT catechin, GA gallic acid, PEG poly(ethylene glycol), MFI mean fluorescence intensity. Source data are provided as a Source Data file.

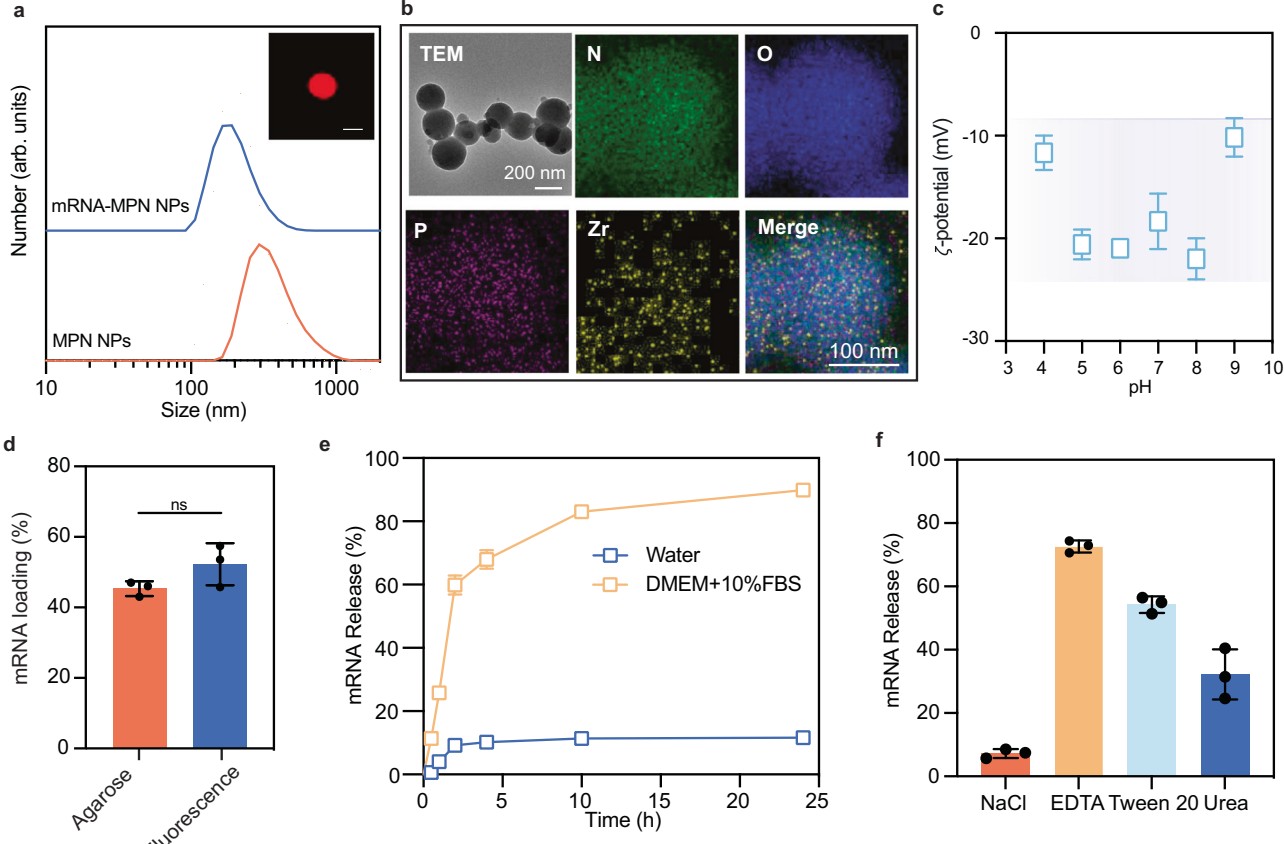

**Fig. 2 | Physicochemical characterization of lead mRNA-MPN NPs. a** Size distribution of mRNA-MPN NPs and MPN NPs measured by dynamic light scattering. The inset shows a Lattice-SIM image of an mRNA-MPN NP using Cy5-labeled mRNA. The scale bar is 200 nm. **b** Transmission electron microscopy (TEM) image and EDX mapping of mRNA-MPN NPs. **c** ζ-Potential of mRNA-MPN NPs across pH 4–9, as measured on a Zetasizer. **d** Percentage of mRNA loading into MPN NPs, as determined by agarose gel electrophoresis and fluorescence spectroscopy. Statistical significance was analyzed by the two-tailed unpaired $t$-test. ns not significant; $p$ (Agarose vs Fluorescence) = 0.1109. **e** Release profiles of mRNA in RNase-free water and DMEM + 10% FBS at 37 °C. **f** Percentage of mRNA released from mRNA-MPN NPs after incubation in different solutions for 24 h. All experiments were performed in triplicates ($n$ = 3), and data are presented as the mean ± SD. The lead mRNA-MPN NPs were assembled with 20k liner PEG, mRNA, EGCG, and Zr$^{IV}$ at a mass ratio of 100:1:100:2.5. DMEM Dulbecco's modified eagle medium, FBS fetal bovine serum, EDTA ethylenediaminetetraacetic acid. Source data are provided as a Source Data file.

composition of mRNA-MPN NPs on the basis of its superior in vitro performance for the subsequent mRNA delivery studies.

## Physicochemical properties of mRNA-MPN NPs

The physicochemical properties of mRNA-MPN NPs are key to their function. The lead NP formulation displayed a spherical morphology with an average diameter of ~220 nm and a narrow size distribution post purification (via centrifugation), as deduced by a polydispersity index (PDI) of 0.19 (Fig. 2a and Supplementary Table 1) and monodisperse NPs observed by lattice-structured illumination microscopy (Lattice-SIM; Supplementary Fig. 10). Notably, the lead mRNA-MPN NPs were smaller (by ~120 nm) than the NP formulation without mRNA (i.e., MPN NPs), indicating that mRNA incorporation may enhance the overall cross-linking density within the NPs. The uniform distribution of elements O, N, P, and Zr within the NPs, as revealed by energy-dispersive X-ray spectroscopy (EDX) mapping (Fig. 2b), along with the

stretching vibration of −NH$_2$ group in the Fourier transform infrared (FTIR) spectrum of the lead mRNA-MPN NPs (Supplementary Fig. 11)[37], indicate the successful incorporation of mRNA and the metal−organic nature of the NPs. Furthermore, the thickness of the mRNA-MPN NPs in the dried state was ~0.8 nm, indicating the collapsible nature of the NPs (Supplementary Fig. 12)[31]. The ζ-potential of the mRNA-MPN NPs was −20 mV at physiological pH (i.e., pH 7), which is consistent with those observed for other MPN materials[27,29]. Moreover, the mRNA-MPN NPs remained negatively charged (i.e., −25 mV to −10 mV) over a wide pH range (i.e., pH 4−9) (Fig. 2c). This noncationic feature sets the current MPN mRNA delivery platform NPs apart from most existing mRNA delivery platforms that contain cationic components, which have raised some safety concerns regarding inflammation and toxicity[16–18].

To elucidate the assembly and stability of mRNA within the MPN NPs, mRNA labeled with cyanine 5 (Cy5) was incorporated during NP assembly. Fluorescence spectroscopy showed ~50% loading efficiency, confirming the observations on agarose gels (Fig. 2d and Supplementary Fig. 13). The stability of the mRNA-MPN NPs and the kinetics of mRNA release were examined in both Milli-Q water and Dulbecco's Modified Eagle Medium (DMEM) supplemented with 10% fetal bovine serum (FBS). As indicated in Fig. 2e, mRNA release from the mRNA-MPN NPs in water was minimal (<12% over 24 h) and without significant change in NP size (Supplementary Fig. 14), confirming the stability of the NPs in water. In FBS-supplemented DMEM, the size of the NPs initially increased (by 30%) within the first 30 min of incubation before gradually decreasing with time. These changes in particle size were accompanied with an mRNA release of 60% in the first 2 h and ~90% by 24 h. The size change in the biological medium suggests that the mRNA-MPN NPs readily interact with proteins present in the environment, likely due to the strong propensity of polyphenols for protein binding[26,27,38]. This interaction was further supported by incubating the mRNA-MPN NPs in Alexa Fluor (AF) 488-conjugated immunoglobulin G (IgG) for 10 min. The Lattice-SIM results revealed an IgG envelope surrounding the NPs (Supplementary Fig. 15), with a computational estimation of 89 antibodies adsorbed per NP. These findings highlight the potential and amenability of the mRNA-MPN NPs toward surface modification and for active delivery via ligand−receptor interactions.

To understand the dominant interactive forces that stabilize mRNA within MPN NPs, we incubated the mRNA-MPN NPs in different solutions for 24 h and quantified the release of mRNA. Upon exposure of the mRNA-MPN NPs to ethylenediaminetetraacetic acid (EDTA; metal chelator) and Tween 20 (hydrophobic competitor), ~70% and ~50% of mRNA was released from the NPs, respectively (Fig. 2f and Supplementary Fig. 16). Incubation with urea, a hydrogen bond competitor, triggered the release of ~30% of mRNA, underlying the role of hydrogen bonds in the stabilization of the NP structure. However, the presence of sodium chloride (NaCl), serving as an ionic competitor, led to an mRNA release of <10%, indicating a minimal contribution of electrostatic forces to the overall stability of the mRNA within the NPs (Fig. 2f and Supplementary Fig. 16).

## mRNA delivery from mRNA-MPN NPs in vitro

The efficient delivery of biomacromolecular therapeutics, including mRNA, to the intracellular environment is often impeded by endosomal entrapment. To address this, we investigated the capability of mRNA-MPN NPs to facilitate endosomal escape via super-resolution microscopy (Lattice-SIM). Endo/lysosomes were stained with Lyso-Tracker Green, while mRNA was labeled with Cy5. Following incubation of the NPs with HEK 293 T cells for 2 h, we observed a notable colocalization of mRNA within endo/lysosomal compartments. The obtained Pearson's correlation coefficient (PCC) of 0.62 suggests a positive correlation between the mRNA and endo/lysosomal markers (Supplementary Fig. 17). At 6 h post incubation, the extent of colocalization decreased, with an accompanying reduction in PCC to 0.45 (Supplementary Fig. 17). At 24 h post incubation, most of the red and

green fluorescence signals appeared as separate signals in the merged SIM image and the color scatter plot, and a corresponding low PCC value (i.e., 0.27) was observed (Fig. 3a). These results collectively suggest a low degree of colocalization of mRNA and endo/lysosomes, indicating endosomal escape of mRNA. The endosomal escape ability of mRNA-MPN NPs is likely attributed to the buffering capacity imparted by MPNs. Although incorporating polyphenols into NPs can facilitate the endosomal escape of mRNA[39,40] (Supplementary Fig. 18), metal−phenolic coordination exerts synergistic buffering effects, enabling superior endosomal escape capability via enhanced proton-sponge effects[41,42] (Supplementary Figs. 17 and 18).

The versatility of mRNA delivery by MPN NPs was further assessed across different mRNA sequences and cell lines in vitro. Given the negligible cytotoxicity of the NPs at concentrations up to 500 ng mRNA per 100 µL media in HEK 293T cells (Supplementary Fig. 19), an intermediate concentration of 166 ng mRNA per 100 µL media was selected for subsequent in vitro transfection studies. The mRNA-MPN NPs were benchmarked against RNAiMax, a commercial cationic lipid-based transfection agent. Transfection was conducted using mRNA encoding firefly luciferase (FLuc). To analyze FLuc transfection, immunofluorescence staining was conducted instead of a typical bioluminescence assay (i.e., based on the reaction between luciferase and luciferin substrate to produce a luminescent product) as EGCG interferes with the assay (Supplementary Fig. 20). The presence of luciferase detected by anti-luciferase antibodies confirmed effective transfection using mRNA-MPN NPs, as shown in Fig. 3b. Expanding beyond reporter proteins, the delivery system was assessed with mRNA encoding a functional protein (i.e., nerve growth factor receptor (NGFR), a member of the tumor necrosis factor (TNF) transmembrane receptor family[43]). The mRNA-MPN NPs mediated surface expression of NGFR with a greater transfection efficiency (by 17.4%) than RNAiMax (Fig. 3c and Supplementary Fig. 21).

A comparative analysis between adherent and suspension cell lines highlighted the efficacy of MPN NPs. In adherent HEK 293T cells, mRNA-MPN NPs enabled nearly 100% transfection efficiency, surpassing RNAiMax by approximately 40%, although with a lower level of protein expression in each cell, as indicated by cell MFI (22,000 for mRNA-MPN NPs vs 40,000 for RNAiMax; Fig. 3d, e). Suspension cells are typically difficult to transfect owing to the limited membrane binding of the NPs[44]. Nevertheless, the mRNA-MPN NPs exhibited superior performance in transfecting Jurkat cells (a suspension T cell line), with both higher transfection efficiency (by 2.5-fold) and average level of protein expression in each cell (i.e., MFI; by ~3-fold) than RNAiMax (Fig. 3g–i). Collectively, these results indicate some degree of structural integrity of the mRNA released from mRNA-MPN NPs and demonstrate the broad applicability of these NPs as an mRNA delivery platform. To improve reproducibility, reporting, and re-analysis, this study conforms to the Minimum Information Reporting in Bio-Nano Experimental Literature (MIRIBEL) standard[45], and a companion checklist is provided in the Supporting Information.

## mRNA expression by mRNA-MPN NPs in vivo

Prior to conducting the in vivo studies, a hemolysis assay was performed to analyze the safety of mRNA-MPN NPs for IV injection. The mRNA-MPN NPs induced < 1% cell lysis across a range of NP concentrations from 10 µg mL$^{-1}$ to 100 µg mL$^{-1}$ (Supplementary Fig. 22), showing the nonhemolytic nature and suitability of the NPs for IV injection. The NPs were then injected intravenously into C57BL/6J mice at an mRNA dosage of 0.25 mg kg$^{-1}$. At 24 h post-injection, major organs, i.e., liver, kidney, lung, heart, spleen, and brain, were harvested for analysis using an in vivo imaging system (IVIS) (Fig. 4a).

For the biodistribution studies, rhodamine 800 (Rh800)-labeled mRNA-MPN NPs were examined. The fluorescently labeled NPs displayed no significant difference in size or ζ-potential (Supplementary Fig. 23 and Supplementary Table 3) from the nonlabeled NPs. As

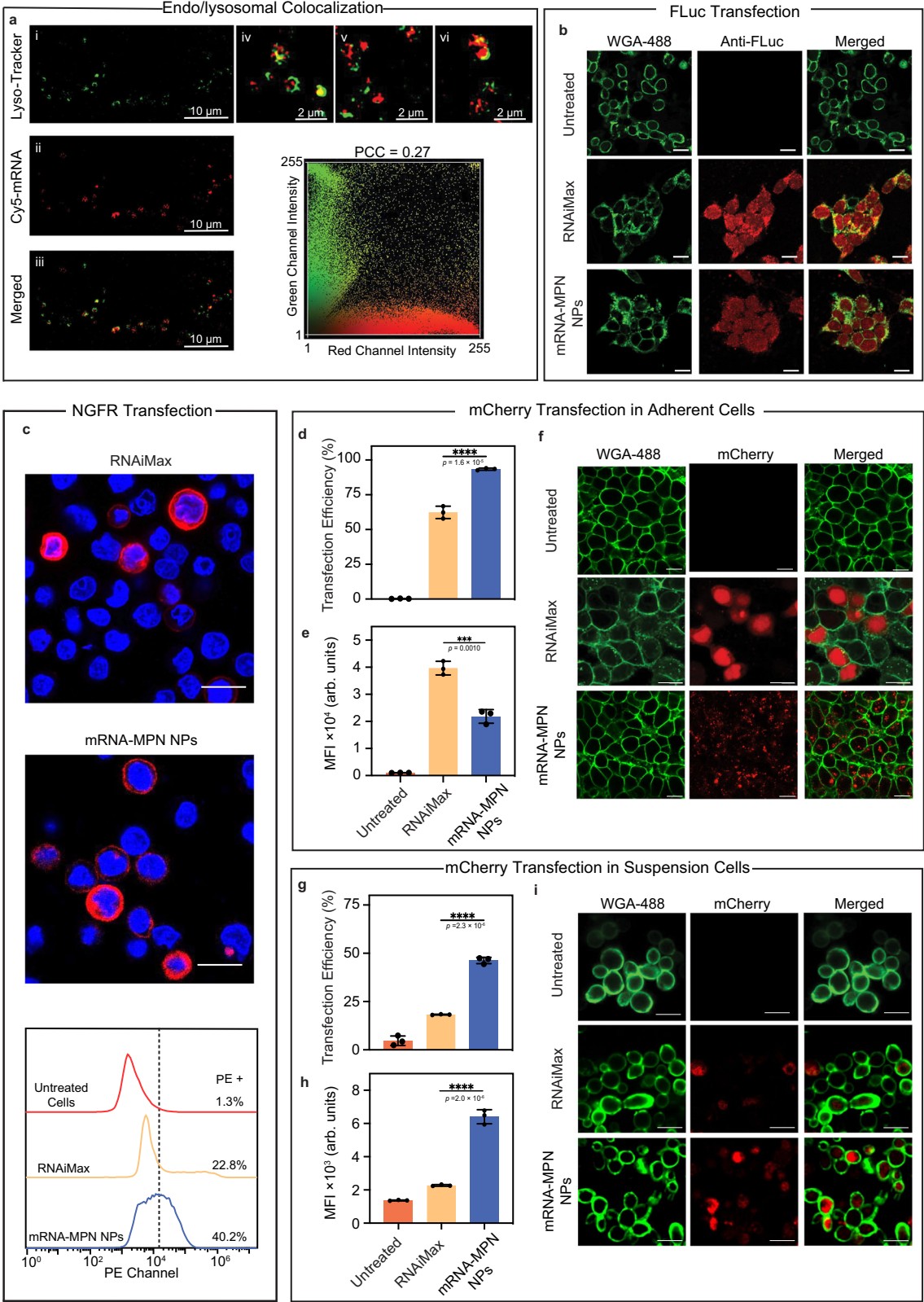

observed from Supplementary Fig. 24, the NPs were broadly distributed in the harvested organs at 4 h, as well as 24 h post-injection. To explore the in vivo mRNA transfection efficacy, an engineered mRNA encoding a brighter red fluorescent protein (mScarlet3)[46] was used, which revealed expression across all harvested organs 24 h after NP injection. Among all organs analyzed, the liver exhibited the highest mScarlet3 expression (~30% of total intensity from all harvested organs; ~13-fold higher radiance efficiency than mice treated with Dulbecco's phosphate-buffered saline (DPBS)) (Fig. 4b and Supplementary Fig. 25). mScarlet3 expression was also predominant in the kidney, and notably, the brain (~20%; 8-fold higher radiance efficiency than in DPBS-treated mice). As the liver exhibited the most prominent level of mRNA expression for mRNA-MPN NPs, the in vivo transfection efficacy of the mRNA-MPN NPs was compared against LNPs formulated

**Fig. 3 | Versatility of in vitro transfection using lead mRNA-MPN NPs. a** (i–iii) Representative SIM images of the intracellular colocalization of mRNA-MPN NPs with endo/lysosomes 24 h post incubation. Endo/lysosomes (green) were stained with LysoTracker Green DND-26. mRNA (red) was conjugated with Cy5. Scale bars are 10 μm. (iv–vi) Representative high-magnification images (scale bars are 2 μm). (vii) Corresponding color scatter plot of endo/lysosomes (green channel) vs mRNA (red channel). **b** CLSM immunofluorescence staining showing the transfection of FLuc-encoding mRNA by mRNA-MPN NPs in HEK 293T cells after 24 h. Cell membranes (green) were stained with AF488-wheat germ agglutinin conjugate (AF488-WGA). Luciferase was detected with AF647-labeled anti-luciferase antibodies (red). Scale bars are 20 μm. **c** Representative CLSM images showing the transfection of NGFR-encoding mRNA in HEK 293T cells after 24 h. NGFR was stained with phycoerythrin-conjugated anti-NGFR. Nuclei were stained with Hoechst 33342. Scale bars are 20 μm. **d–f** Percentage of transfection (**d**), MFI (**e**), and representative

CLSM images (**f**) of adherent cells (HEK 293T) transfected by RNAiMax or mRNA (mCherry)-MPN NPs after 24 h. Cell membranes were stained green. Scale bars are 20 μm. **g–i** Percentage of transfection (**g**), MFI (**h**), and representative CLSM images (**i**) of suspension cells (Jurkat) transfected by RNAiMax or mRNA (mCherry)-MPN NPs after 24 h. Cell membranes were stained green. Scale bars are 20 μm. All experiments were performed in triplicates ($n = 3$) and data are presented as mean values ± SD. Statistical significance was analyzed using one-way ANOVA with Tukey's multiple comparisons test. In **d**, $p$ (RNAiMax vs mRNA-MPN NPs) = $1.6 \times 10^{-5}$. In **e**, $p$ (RNAiMax vs mRNA-MPN NPs) = 0.0010. In **g**, $p$ (RNAiMax vs mRNA-MPN NPs) = $2.3 \times 10^{-6}$. In **h**, $p$ (RNAiMax vs mRNA-MPN NPs) = $2.0 \times 10^{-6}$. The lead mRNA-MPN NPs were assembled with 20k liner PEG, mRNA, EGCG, and Zr$^{IV}$ at a mass ratio of 100:1:100:2.5. Cy5 cyanine 5, FLuc firefly luciferase, NGFR nerve growth factor receptor, MFI mean fluorescence intensity. Source data are provided as a Source Data file.

with SM-102 ionizable lipid (used in Moderna's SARS-CoV-2 vaccine vector, which show predominant liver expression). The results showed that the mRNA-MPN NPs enabled 2-fold higher expression in the liver, as well as overall higher expression across all harvested organs relative to the mRNA-SM-102 LNPs at the same mRNA dosage (Fig. 4c, d and Supplementary Figs. 26 and 27).

To further validate mRNA expression in the organs using mRNA-MPN NPs, sectioned organ analysis was performed. In this analysis, mRNA encoding a far-red fluorescent protein (i.e., emiRFP670)[47] was employed to alleviate the interference from autofluorescence in tissues (emission occurs at a longer wavelength where there is less tissue autofluorescence). The emiRFP670-encoding mRNA (1232 nucleotides) had a similar nucleotide number to mScarlet3-encoding mRNA (1004 nucleotides) (Supplementary Table 2). The mRNA (emiRFP670)-MPN NPs and mRNA (mScarlet3)-MPN NPs displayed comparable particle size (Supplementary Fig. 23 and Supplementary Table 1) and comparable in vivo transfection (Supplementary Fig. 28a–d). CLSM images revealed the presence of emiRFP670-positive cells in the liver, kidney, and brain tissues (Fig. 4e–g and Supplementary Fig. 28e–j), consistent with the organ-level mRNA expression.

Next, we examined the ability of the mRNA-MPN NPs to mediate gene editing by employing genetically engineered tdTomato (tdTom) reporter mice and Cre recombinase (Cre) mRNA. In these reporter mice, tdTom expression is silenced by a loxP-flanked stop cassette, which can be removed by the Cre enzyme to activate fluorescence (Fig. 4h)[48]. Successful gene editing across the liver, kidney, and brain was indicated by the presence of tdTom signals (Fig. 4i, j). Upon staining the vasculature with lectin, some of the tdTom signals did not colocalize with the cerebral vascular structures, indicating transfection beyond the blood–brain barrier. This brain deposition may primarily be mediated by the composition of the mRNA-MPN NPs. Based on the compositional mass ratio of the lead NP formulation (i.e., PEG:mRNA:EGCG:Zr$^{IV}$ = 100:1:100:2.5), EGCG is one of the most predominant constituents in mRNA-MPN NPs. EGCG has been shown to transiently enhance the blood–brain barrier permeability[49–51], which may provide a possible mechanism for the entry of mRNA-MPN NPs into the brain. These findings present exciting opportunities to deliver mRNA to the brain, which, to our knowledge, has remained elusive with current mRNA delivery systems. Although further work is required to identify the cell types transfected in the brain, brain deposition enabled by mRNA-MPN NPs opens up avenues for the development of therapeutics for the possible prevention and treatment of brain tumors and neurological disorders (e.g., Parkinson's disease, Alzheimer's disease). Moreover, studies have shown that delivering polyphenols to the brain can modulate microglia-mediated inflammation and exert neuroprotective effects[52,53], hence providing further motivation for the application of mRNA-MPN NPs in the brain.

## In vivo safety and biocompatibility of mRNA-MPN NPs

Comprehensive toxicity and biocompatibility evaluations post-IV administration were conducted. Cytokine (interleukin-6 (IL-6) and TNF-α) release assays in the blood (Fig. 5a) and tissue histological analyses (i.e., hematoxylin and eosin (H&E) staining, Fig. 5b) revealed no significant inflammatory response, cellular swelling, or histological alterations, denoting minimal in vivo toxicity of the NPs within the 24 h time period examined. Furthermore, mice exhibited negligible weight loss within 10 days post-IV administration (Supplementary Fig. 29). To assess any potential toxicity of the metal ions in the NPs, the biodistribution of Zr in organs was assessed up to 10 days post-administration using inductively coupled plasma mass spectrometry (ICP-MS) on digested organ and body fluid (i.e., urine and blood) preparations. The mRNA-MPN NPs contained a Zr load of 64%, implying that for every 0.25 mg kg$^{-1}$ of mRNA delivered to mice, approximately 0.8 mg kg$^{-1}$ of metal will be introduced. However, under experimental conditions, NPs resulted in only nanogram-level changes in Zr content per gram of organ or body fluid (Fig. 5c and Supplementary Fig. 30). At 4 h post-injection, approximately 60 ng of Zr was detected per gram of blood, which was the highest level among all tested time points within 10 days (Fig. 5c), suggesting the presence of Zr in circulation within 4 h. The gradual reduction of Zr content in the blood after 4 h indicates gradual organ deposition or excretion. In the urine samples, ~70 ng of Zr was detected per gram of urine at 4 h, which decreased rapidly thereafter (Fig. 5c and Supplementary Fig. 30), suggesting efficient metal excretion from mice. Although Zr signals in harvested organs peaked on either day 3 (e.g., liver, kidney) or day 6 (e.g., brain, lung), they reduced to the baseline level (i.e., Zr content in DPBS-treated mice), implying that the overall Zr accumulation owing to the NPs was negligible within the 10-day study. The dose-dependent effect on Zr accumulation was also evaluated by applying a second injection on day 5. The Zr content slightly increased in the urine, blood, and most harvested organs on day 6 and was then excreted by day 10 (Supplementary Fig. 30). Overall, the percentage of the injected dose of Zr per gram tissue (%ID g$^{-1}$) in harvested organs and body fluids was low (i.e., <2%) regardless of single or double injections, further implying minimal in vivo metal accumulation. Together these data highlight the safety and biocompatibility of the mRNA-MPN NPs.

## Organ tropism of mRNA expression modulated by NP composition

The effects of mRNA dosage, treatment time, and NP composition on in vivo mRNA transfection were examined. Unlike a dosage of 5 μg of mRNA per mouse (~0.25 mg kg$^{-1}$) that led to prominent expression in the liver, kidney, and brain, using a lower dosage of 2 μg per mouse (~0.1 mg kg$^{-1}$) was insufficient to trigger expression that could be detected via IVIS. In contrast, increasing the dosage to 0.5 mg kg$^{-1}$ only resulted in a comparable level of mRNA expression to that observed

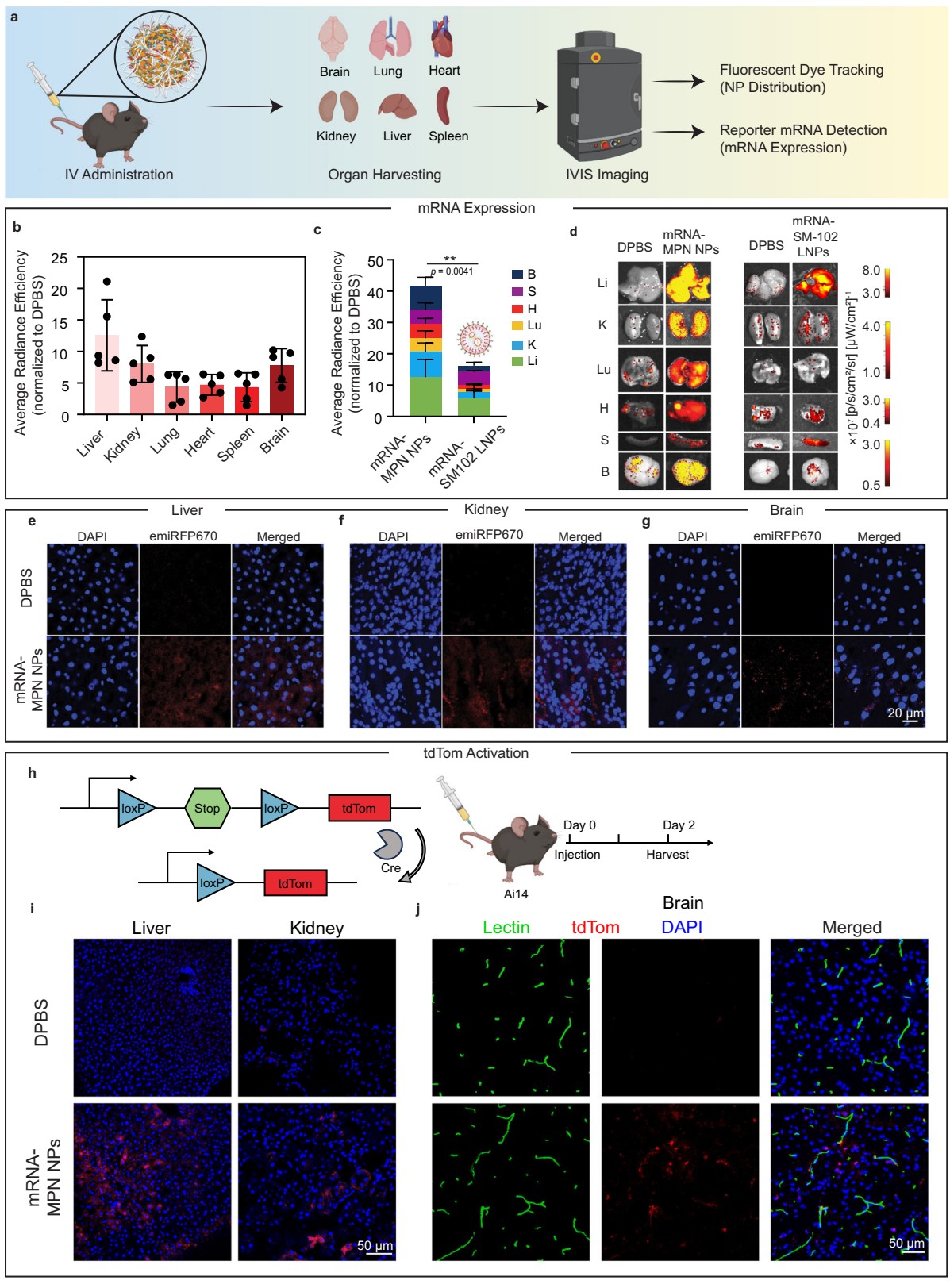

using 0.25 mg kg⁻¹ (Supplementary Fig. 31). The expression of mScarlet3 mRNA was negligible at 12 h post-IV administration and peaked at 24 h (Fig. 6a, b Supplementary Fig. 32, with the intensity of mRNA expression illustrated as normalized radiance efficiency). Overall, the mRNA expression level decreased at 48 h, although mRNA expression remained strong in the spleen, likely owing to the predominant splenic retention of the NPs after 48 h (Supplementary Fig. 33).

Leveraging the high degree of modularity of MPN NPs, we investigated the effect of varying mRNA-MPN NP constituents on in vivo mRNA expression. Altering the Zr^IV-to-EGCG mass ratio influenced both mRNA expression levels and the percentage of mRNA expression in the harvested organs (Supplementary Fig. 34). For example, using mRNA-MPN NPs without metal ions (i.e., Zr^IV-to-EGCG mass ratio = 0:40) led to a negligible mScarlet3 fluorescence signal or expression in

**Fig. 4 | In vivo mRNA transfection in C57BL/6J mice using mRNA-MPN NPs.**
**a** Schematic illustration of IV administration of mRNA-MPN NPs and subsequent IVIS imaging. **b** mScarlet3 expression in different organs of C57BL/6J mice analyzed by IVIS 24 h post IV injection of mRNA-MPN NPs (lead formulation; mRNA at a dose of 0.25 mg kg⁻¹). **c**, **d** Comparison of mScarlet3 expression using mRNA-MPN NPs and mRNA-SM-102 LNPs: both quantitative (**c**) and representative images (**d**) were obtained from IVIS. For **b**, **c** five mice were included in each group (*n* = 5), and the quantitative data were normalized to DPBS-treated mice (negative control) and presented as mean values ± SD. The fluorescence signal observed in DPBS-treated mice may be due to tissue autofluorescence, which is often observed at the excitation and emission wavelengths of 561 nm and 594 nm, respectively. Statistical significance was analyzed using the two-tailed unpaired *t*-test. In **c**, *p* (mRNA-MPN NPs vs mRNA-SM102 LNPs) = 0.0041. **e**–**g** CLSM images of sectioned liver (**e**),

kidney (**f**), and brain (**g**) tissues post IVIS imaging. Cell nuclei were stained with 4′,6-diamidino-2-phenylindole (DAPI). Red fluorescence indicates expressed emiRFP670. **h** Schematic illustration of an Ai14 Cre reporter model that could express tdTom by translating Cre-recombinase mRNA to Cre protein to delete the loxP-flanked stop cassette. **i**, **j** CLSM images of sectioned liver and kidney (**i**) and brain (**j**) tissues 48 h post IV injection of Cre-MPN NPs (mRNA dose at 0.25 mg kg⁻¹). Red fluorescence indicates expressed tdTom. Cell nuclei were stained with DAPI. Blood vessels were stained with Lectin-Dylight 488, which was injected intravenously 5 min before euthanizing the mice. IV intravenous, IVIS in vivo imaging system, DPBS Dulbecco's phosphate-buffered saline, LNPs lipid nanoparticles, tdTom tdTomato. Panel (**a**) and parts of panels (**c**, **h**) were created with BioRender.com. Source data are provided as a Source Data file.

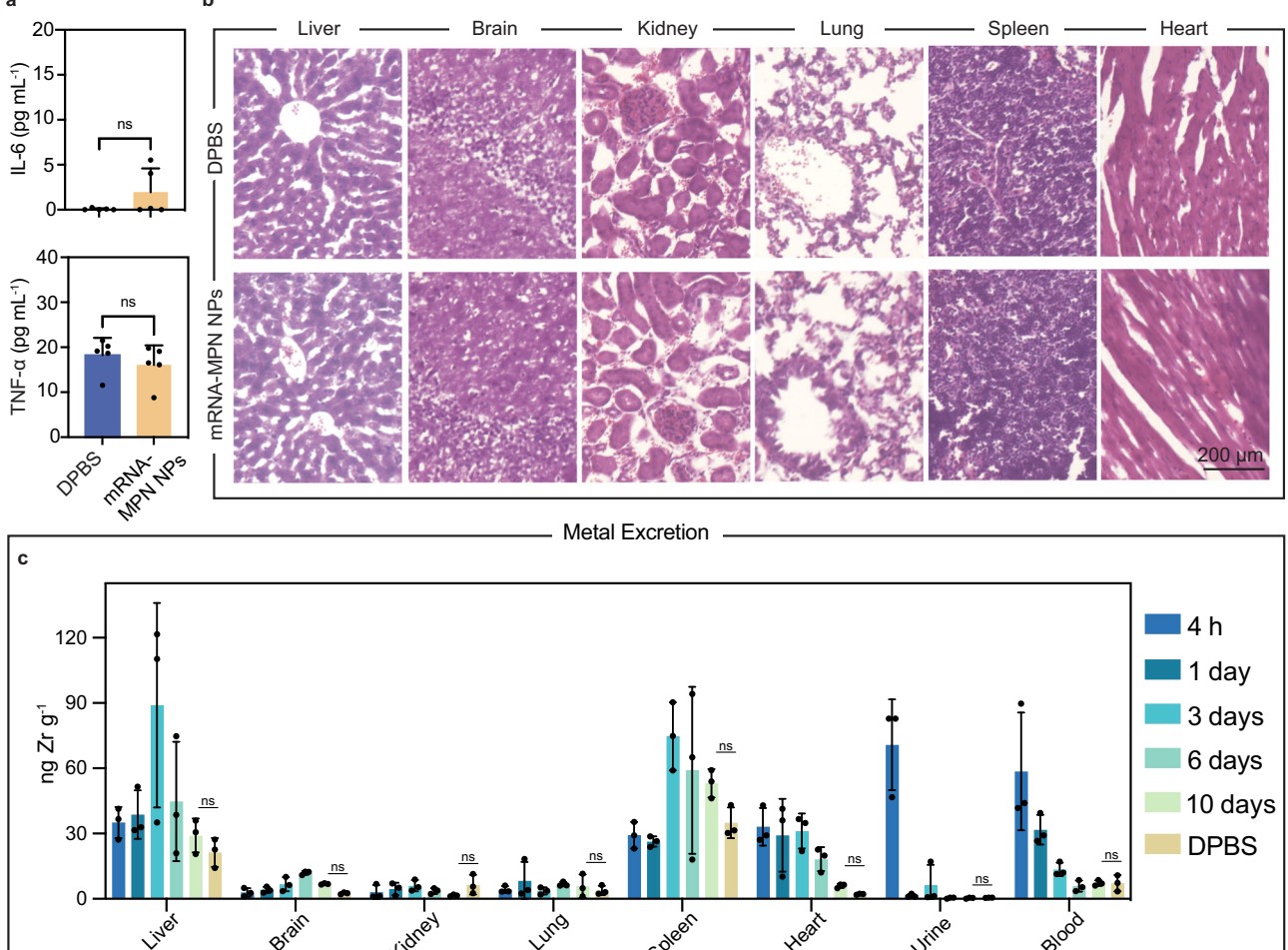

**Fig. 5 | In vivo biocompatibility and metal excretion from mRNA-MPN NPs.**
**a** Release of cytokines (IL-6 and TNF-α) in plasma collected from mice treated with DPBS (negative control) or mRNA-MPN NPs. Statistical significance was analyzed using the two-tailed unpaired *t*-test. *p* (DPBS vs mRNA-MPN NPs): IL-6, 0.1514; TNF-α, 0.4337. The plasma from five biologically independent mice was included in each group (*n* = 5) and the data are presented as mean values ± SD. **b** H&E staining of organs harvested from mice treated with DPBS or mRNA-MPN NPs after 24 h. **c** Zr excretion profiles from harvested organs and body fluids (i.e., urine and blood) post-treatment with DPBS or mRNA-MPN NPs, as measured by ICP-MS elemental

analysis. Zr excretion is presented as the amount of Zr per gram of organ or body fluid collected from mice. Statistical significance was analyzed using two-way ANOVA. *p* (DPBS vs 10 days): liver, 0.6050; brain, 0.9979; kidney, 0.9959; lung, >0.9999; spleen, 0.4403; heart, 0.9988; urine, >0.9999; and blood, >0.9999. Three biologically independent mice were included for each group (i.e., each time point and DPBS) (*n* = 3). Data are presented as mean values ± SD. IL-6 interleukin-6, TNF-α tumor necrosis factor-α, DPBS Dulbecco's phosphate-buffered saline. Source data are provided as a Source Data file.

most harvested organs, suggesting the vital role of metal ions in improving the overall level of mRNA transfection. However, this ratio resulted in a higher proportion of mScarlet3 expression in the brain (a 4.5-fold increase in radiance efficiency compared to the DPBS-treated controls) than 1:40 and 3:40 ratios. Assembling mRNA-MPN NPs at a

Zr^IV-to-EGCG mass ratio 1:40 ratio (i.e., lead formulation) led to the most pronounced mScarlet3 expression in the liver (Fig. 6c) among the different Zr^IV-to-EGCG mass ratios examined. Further increasing the Zr^IV-to-EGCG mass ratio improved the proportion of mScarlet3 expression in the kidney, while reducing the proportion of the

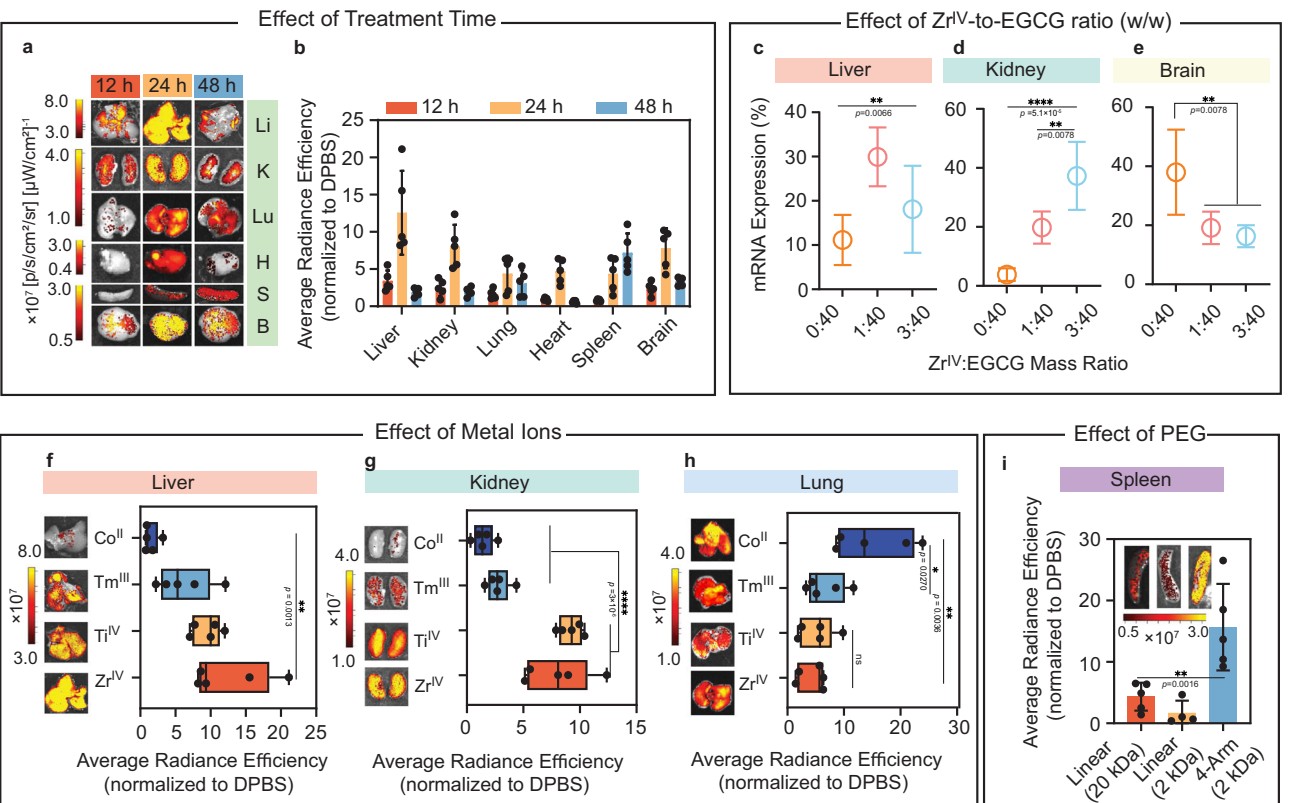

**Fig. 6 | In vivo mRNA expression in C57BL/6J mice using mRNA-MPN NPs with various highly performing formulations. a–i** mScarlet3 expression in harvested organs using MPN NPs under varying conditions: treatment time (**a**, **b**); $Zr^{IV}$-to-EGCG mass ratio (**c–e**); metal ions (**f–h**); and $M_w$ and structure of PEG (**i**). Both quantitative results (i.e., bar diagrams) and representative images of the biodistribution were obtained by IVIS. For all quantitative results demonstrated in Fig. 6, four or five biologically independent mice were included in each group (n = 4 or 5), and the quantitative data were normalized to DPBS (negative control). In **f–h** the whiskers of the plot define the minimal and maximal values of individual data points, with the center line of the box representing the mean value of the data set. Li liver, K kidney, Lu lung, H heart, S spleen, B brain, PEG poly(ethylene glycol), EGCG epigallocatechin. Statistical significance was analyzed using one-way ANOVA or one-way ANOVA with Tukey's multiple comparisons test. In **c** \*\*$p$ = 0.0066. In **d** \*\*\*\*$p$ = 5.1 × 10⁻⁵, \*\*$p$ = 0.0078. In **e** \*\*$p$ = 0.0078. In **f** \*\*$p$ = 0.0013. In **g** \*\*\*\*$p$ = 3 × 10⁻⁶. In **h** \*\*$p$ = 0.0036, \*$p$ = 0.0270. In **i** \*\*$p$ = 0.0016. All data are presented as mean values ± SD. DPBS Dulbecco's phosphate-buffered saline, EGCG epigallocatechin, PEG poly(ethylene glycol). Source data are provided as a Source Data file.

expression in the brain (Fig. 6d, e, Supplementary Figs. 34 and 37, and Supplementary Table 4).

The modularity and versatility of the mRNA-MPN NPs were further demonstrated by varying the metal ion component. Similar to using $Zr^{IV}$, using $Co^{II}$, $Tm^{III}$, or $Ti^{IV}$ enabled successful mRNA transfection in vivo (Supplementary Figs. 35 and 37 and Supplementary Table 5). The type of metal ions influenced the organ tropism of expressed mRNA (Fig. 6f–h). Replacing $Zr^{IV}$ with $Co^{II}$ resulted in <5% mRNA expression in the liver and kidney but ~50% expression in the lung (i.e., ~50%, 13-fold higher radiance efficiency than DPBS-treated mice). In contrast, substituting $Zr^{IV}$ with $Ti^{IV}$ led to a comparable expression profile in the liver (i.e., ~30%, ~10-fold higher radiance efficiency than DPBS-treated mice), kidney (i.e., ~20%, ~9-fold higher radiance efficiency than DPBS-treated mice), and lung (i.e., ~11%, 4.5-fold higher radiance efficiency than DPBS-treated mice). Varying the type of metal ions leads to compositional changes on mRNA-MPN NPs, potentially altering the composition of the biomolecular corona formed on the NP surface, as shown previously with MPN-based NPs[54]. This variability in corona composition may influence the organ tropism of mRNA expression, which was also observed with LNPs[55].

The mRNA expression could also be adjusted by using different PEGs. Assembling mRNA-MPN NPs using 2k linear PEG led to a reduced level of overall mRNA expression compared to using 20k linear PEG but exhibited liver tropism. Using 2k 4-arm PEG also resulted in an overall diminished level of mRNA expression compared to using 20k

linear PEG (Supplementary Fig. 36), but the splenic expression increased from 10% to 50% with the normalized radiance efficiency increasing from 4-fold to 16-fold (normalized to DPBS-treated mice) (Fig. 6i and Supplementary Fig. 36). The increment of splenic expression might be attributed to the change in ζ-potential. NPs prepared with 2k 4-arm PEG displayed more negative ζ-potential (−47 ± 5 mV) than those prepared with the linear PEG (−17 ± 4 mV; Supplementary Fig. 37 and Supplementary Table 6), which might increase the likelihood of splenic deposition of NPs[56]. In addition, protein adsorption (i.e., the formation of a protein corona) on the NPs may play a role in driving organ tropism, as demonstrated with other NP systems[55].

In summary, we developed a modular and noncationic metal–organic NP platform (i.e., mRNA-MPN NPs) for both in vitro and in vivo mRNA delivery. These NPs, approximately 200 nm in size and negatively charged, owe their versatility to the universal binding affinity of phenolic ligands, facilitating the robust integration of mRNA with diverse sequences. The mRNA-MPN NPs demonstrated superior transfection efficiency in vitro over a commercial transfection agent and a comparable in vivo transfection level to LNPs without causing significant toxicity or inflammation. Protein expression was primarily observed in the liver, kidney, and brain after IV administration of lead mRNA-MPN NPs (composed of 20 kDa linear PEG, selected mRNA, EGCG, and $Zr^{IV}$ at a mass ratio of 100:1:100:2.5). Notably, the modularity of this platform afforded a diverse choice of building blocks for NP engineering, which could readily alter organ tropism of mRNA

transfection. This modular metal–organic nature may enable further engineering of MPN NPs through superstructure formation and surface modification, facilitating the rational design of delivery vehicles for a wide array of therapeutics targeted to specific tissues. Therefore, we envision that the NP platform reported herein will further advance NP-mediated therapeutics, widening the breadth of diseases that can be addressed with mRNA NP technologies.

## Methods

### Ethics

This work was conducted in accordance with the Australian code for the care and use of animals for scientific purposes, and experiments were approved by the University of Melbourne Animal Ethics Committee (Ethics 10404 and 27608).

### Materials

DPBS (catalog number #14190144), ultrapure DNase-/RNase-free distilled water (#10977015), DMEM (#10569069), reduced serum medium (Opti-MEM; #31985088), Roswell Park Memorial Institute (RPMI)-1640 medium (#21870092), Lectin-Dylight 488 (#L32470), and 1,1'-dioctadecyl-3,3,3',3'-tetramethylindotricarbocyanine iodide (DiR; #D12731) were purchased from Thermo Fisher Scientific. EGCG (# E4143), TA (#403040), CAT (#G6657), GA (#91215), NaCl (#S9888), copper(II) chloride dihydrate (CuCl$_2$·2H$_2$O; #307483), cobalt(II) chloride (CoCl$_2$; #232696), thulium(III) chloride hexahydrate (TmCl$_3$·6H$_2$O; #204668), chromium(III) chloride hexahydrate (CrCl$_3$·6H$_2$O; #230723), titanium(IV) bis(ammonium lactato)dihydroxide (#388165), zirconium(IV) chloride (ZrCl$_4$; #221880), PEG (2k linear #821037, 20k linear #81275), agarose (low EEO; #A0576), EDTA (# E8008), urea (#U5378), and Tween 20 (#P7949) were purchased from Sigma-Aldrich. PEG (2k 4-arm, #4ARM-PEG) was purchased from JenKem (USA). 8-[(2-Hydroxyethyl)[6-oxo-6-(undecyloxy)hexyl]amino]-octanoic acid, 1-octylnonyl ester (SM-102; #C1042) was purchased from APExBIO (TX, USA). 1,2-Distearoyl-sn-glycero-3-phosphocholine (DSPC; #850365), cholesterol (#700100), and 1,2-dimyristoyl-rac-glycero-3-methoxypolyethylene glycol-2000 (DMG-PEG 2000; #880151) were purchased from Avanti Polar Lipids (AL, USA). mRNA encoding FLuc (#FLUC1000P), mCherry (#MCHE1000P), NGFR (#NGFR500P), mScarlet3 (#FLAME1000P), emiRFP670 (#NIRFP1000P), or Cre recombinase (#CRE1000P) and Cy5-labeled mRNA (#EGFP100P-Cy5) were purchased from Messenger Bio Pty Ltd. (Australia).

### Fabrication of mRNA-MPN NPs

The lead formulation was prepared as follows: 20k linear PEG (20 µL, 10 mg mL$^{-1}$ in RNase-free water) and mRNA (2 µL, 1 mg mL$^{-1}$ in citric buffer) were first mixed in a 1.7 mL Eppendorf tube. EGCG (40 µL, 5 mg mL$^{-1}$ in RNase-free water) was then introduced to initiate assembly, and the NPs were further stabilized by pipetting in ZrCl$_4$ (10 µL, 0.5 mg mL$^{-1}$ in RNase-free water). The mixture was then kept on ice for 30 min. For in vitro mRNA transfection during the formulation screen, the cocktail solution was directly added to cells without purification.

To prepare mRNA-MPN NPs incorporating different phenolic building blocks (i.e., TA, EGCG, CAT, or GA), PEG (i.e., 2k linear, 20k linear, or 2k 4-arm), and metal ions (i.e., Na$^I$, Cu$^{II}$, Co$^{II}$, Tm$^{III}$, Cr$^{III}$, Zr$^{IV}$, or Ti$^{IV}$), the mass ratio of each component was fixed as above (PEG:mRNA:phenolic ligand:metal ion = 100:1:100:2.5) unless otherwise specified.

### Characterization of mRNA-MPN NPs

For mRNA-MPN NP characterization, and in vitro and in vivo studies, particles were purified by centrifugation (8000×$g$, 5 min). The resulting pellet was dispersed in RNase-free water (100 µL), followed by sonication for 10 s to fully resuspend the NPs.

To determine the mRNA loading capacity of mRNA-MPN NPs, the mixture of PEG, mRNA, phenolic ligand, and metal ion was diluted with RNase-free water and loaded on 1% agarose gels with a concentration of 200 ng mRNA per well. Note that the uncropped unprocessed gel scan for the inset of Fig. 1b with an RNA marker is included in the Source Data file. A Zetasizer Nano-ZS instrument (Malvern Instrument, UK) was used to measure the hydrodynamic diameter (number mean), PDI, and ζ-potential of mRNA-MPN NPs in different buffered pH buffers (10 mM). UV-Vis absorption spectra were recorded on a Specord 250 Plus spectrophotometer (Analytik Jena AG, Germany). FTIR spectroscopy analysis was performed on a Tensor II FTIR spectrometer (Bruker Optics, USA). Super-resolution fluorescence images of the particles in the aqueous phase were obtained on a structured illumination microscope (Elyra 7 Lattice SIM with Lattice SIM$^2$ processing function, Zeiss). TEM measurements were performed on an FEI Tecnai F20 microscope (FEI Company, Hillsboro, OR, USA) at an operating voltage of 200 kV to obtain TEM images, high-angle annular dark-field images, and EDX mapping data. Atomic force microscopy (AFM) experiments were performed on a JPK NanoWizard II BioAFM instrument (JPK Instruments AG, Berlin, Germany) with tapping-mode cantilevers.

### In vitro release studies

To study the kinetics of mRNA release, mRNA-MPN NPs with Cy5-labeled mRNA were incubated in different media (RNase-free water, DPBS, and DPBS with 10% FBS) at 37 °C. At each different time point, the particles were centrifuged at 8000×$g$ for 5 min. The supernatant was removed for fluorescence measurements and replaced with the same amount of fresh medium for the next time point. The fluorescence of the collected supernatant was measured using an Infinite M200 microplate reader (Tecan, Switzerland), and the concentration of Cy5-labeled mRNA in the supernatant was calculated using a standard curve (Supplementary Fig. 13).

### Disassembly assay

To determine the dominant interactive forces for mRNA-MPN NP assembly, the particles were incubated in NaCl, EDTA, Tween 20, and urea (100 mM). After 24 h, mRNA-MPN NPs in different solutions were loaded into 1% agarose gels. The gels were run at 90 V for 45 min. Bands of naked mRNA and untreated mRNA-MPN NPs were set to to 100% and 0% disassembly, respectively. Band intensity was obtained and analyzed by Image J software.

### Antibody adsorption

To determine the antibody adsorption capacity on the surface of mRNA-MPN NPs, AF488-conjugated IgG (1 µL, 2 mg mL$^{-1}$, Thermo Fisher Scientific, #A-11008) was mixed with the mRNA-MPN NPs and incubated for 10 min. Subsequently, the mixture was centrifuged at 8000×$g$ for 5 min to wash off the excess IgG. The IgG fluorescence of the supernatant was compared to the initial fluorescence using an Infinite M200 microplate reader to calculate the loading amount and total number of IgG adsorbed on the NP surface. Nanoparticle tracking analysis was conducted to measure particle concentration on a NanoSight NS300 (Malvern Panalytical, UK). This allowed for the calculation of the number of antibodies adsorbed onto each particle. The synthesized NPs were visualized using Lattice-SIM.

### Cell culture

HEK 293T cells with low passage numbers (passages: 14–25) were cultured in DMEM supplemented with 10% ($v/v$) FBS and 1% ($v/v$) penicillin/streptomycin. Jurkat cells (passages: 11–15) were cultured in supplemented RPMI-1640. Cells were grown at 37 °C, 5% CO$_2$, and 95% humidity.

### In vitro transfection

Two reporter mRNAs FLuc and mCherry and a functional mRNA NGFR were used to evaluate the mRNA transfection and demonstrate the versatility of transfection with mRNA-MPN NPs. HEK 293T cells were

seeded at a density of $6 \times 10^4$ cells per well in 48-well plates or in 8-well Lab-Tek chamber slides unless otherwise specified. Jurkat cells were seeded at a density of $3 \times 10^4$ cells per well in 96-well round-bottom plates. After overnight incubation (80–90% HEK 293T confluency), the cell media was aspirated and an aliquot (300 μL) of mRNA-MPN NPs in serum-free medium (Opti-MEM) was added to achieve a final mRNA concentration of 500 ng per well. An equivalent concentration of mRNA was mixed with the commercial transfection reagent (Lipofectamine, RNAiMax) according to the product manual as a positive control. Naked mRNA (equivalent concentration) was also set as a control. After 6 h of incubation, the serum-free medium was aspirated and replaced with complete DMEM (RPMI-1640 for Jurkat). Incubation was continued for another 18 h before the cells were washed thrice with DPBS, trypsinized, and transferred into a 96-well plate for further washing and flow cytometry.

For CLSM imaging following mCherry mRNA transfection, cells were seeded in 8-well Lab-Tek chamber slides with the other conditions kept the same. After transfection, cells were fixed with 4% paraformaldehyde (PFA; Thermo Fisher Scientific, #J19943.K2) (in DPBS) for 10 min, followed by staining of the cell membranes with AF488-WGA (5 μg mL$^{-1}$; Thermo Fisher Scientific, #W11261) and cell nuclei with Hoechst 33342 (5 μg mL$^{-1}$; Thermo Fisher Scientific, #62249).

For CLSM imaging following FLuc mRNA transfection, cells were seeded in 8-well Lab-Tek chamber slides with the other conditions kept the same. After transfection, cells were fixed with 4% PFA for 10 min. Then, the cell membranes were permeabilized using 0.2% Triton-X for 10 min, followed by blocking with 2% bovine serum albumin (BSA) for 1 h at room temperature. The cells were incubated with a rabbit anti-Firefly luciferase (1/500 dilution, Abcam, #ab232629) at 4 °C overnight. After washing, the AF647-conjugated secondary antibody (4 μg mL$^{-1}$, Thermo Fisher Scientific, #A-21244) was added and incubation was conducted for 1 h at room temperature, followed by three rounds of washing with DPBS.

For flow cytometry and confocal imaging following NGFR mRNA transfection, cells were seeded in 24-well plates at a density of $15 \times 10^4$ cells per well with the other conditions kept the same. After transfection, cells were washed thrice with DPBS, trypsinized, and transferred into a 96-well plate for washing by centrifugation. The cells were then fixed with 4% PFA for 10 min followed by blocking with 2% BSA for 1 h at room temperature. The cells were then treated with mouse anti-NGFR (2.5 ng μL$^{-1}$ according to the manufacturer's instructions, eBioscience, #12-9400-42) overnight at 4 °C and washed thrice with DPBS before running flow cytometry and confocal imaging experiments. For SIM imaging following NGFR mRNA transfection, cells were seeded in 8-well Lab-Tek chamber slides with the other culturing and treatment conditions kept the same.

For flow cytometry analysis, data were collected using a BD Accuri C6 Plus and then processed using FlowJo software (V.10). Fragments and debris were gated out from monoculture cell lines through the scatter plots (FSC-H vs SSC-H). Data points beyond the fluorescence intensity histogram of untreated cells were recognized as "positive". Gating strategies for the flow cytometry are provided in Supplementary Fig. 38.

## Endo/lysosomal colocalization

HEK 293T cells were seeded at a density of $6 \times 10^4$ cells per well in 8-well Lab-Tek chamber slides and cultured overnight to allow for cellular adhesion to the substrate. The following day, the cell medium was aspirated and an aliquot (300 μL) of mRNA-MPN NPs prepared with Cy5-labeled mRNA in Opti-MEM was added to achieve a final mRNA concentration of 1 μg per well. For the 2 h and 6 h time points, Opti-MEM has aspirated 2 h or 6 h post-NP treatment and replaced with DMEM containing 75 nM LysoTracker Green DND-26 (Thermo Fisher Scientific, #L7526) for further incubation for 1 h at 37 °C to stain endo/lysosomes, followed by washing thrice with DPBS. For the 24-h

time point, DMEM supplemented with 10% FBS was introduced 6 h post-treatment and left for 18 h before staining with LysoTracker Green and washing. The colocalization of endo/lysosomes with mRNA was observed via SIM with a 63× oil immersion objective. The images were further processed by Image J software to obtain PCC values.

## Live/dead staining

HEK 293T cells were seeded in 96-well plates at a density of $2 \times 10^4$ cells per 100 μL complete DMEM and cultured overnight. mRNA-MPN NPs in Opti-MEM with a gradient mRNA concentration (10 ng, 50 ng, 100 ng, 200 ng, 300 ng, and 500 ng per well) were added for incubation. After 6 h, Opti-MEM was aspirated and replaced with complete DMEM. The aspirated media of each well with floating cells were collected and incubated in a new plate. After 18 h, complete DMEM was aspirated, and the media with floating cells were collected. All the removed cells were transferred to their original well before washing thrice with DPBS and incubation with live/dead fixable far-red stain (Thermo Fisher Scientific, #L34974) at 37 °C for 30 min. The viability of the cells was then analyzed with flow cytometry.

## Hemolysis analysis

Whole blood from C57BL/6J mice (female, 8–10-weeks-old) was collected in anticoagulant tubes and centrifuged at 3500×$g$ for 5 min to isolate red blood cells (RBCs) from plasma. The RBCs were then washed thrice with DPBS. Then, the supernatant was discarded, and DPBS was added to obtain a 10% RBC suspension. mRNA-MPN NPs with different concentrations (10 μg mL$^{-1}$, 20 μg mL$^{-1}$, 50 μg mL$^{-1}$, 100 μg mL$^{-1}$) were added to attain a final RBC concentration of 2% for subsequent incubation for 2 h, with DPBS as negative control and water as positive control. All samples were then centrifuged at 9000×$g$ for 5 min. An aliquot (100 μL) of the supernatant was transferred to a 96-well plate, and the absorbance of hemoglobin was measured by a microplate reader at 577 nm. The percentage hemolysis of RBCs was calculated using the following equation:

$$\%\text{Hemolysis} = \frac{Ab_{NP} - Ab_{DPBS}}{Ab_{water} - Ab_{DPBS}} \times 100\% \qquad (1)$$

where $Ab_{NP}$ is the absorbance readout from blood treated with NPs, $Ab_{DPBS}$ is the absorbance readout from blood in DPBS and $Ab_{water}$ is the absorbance from blood in water.

## Simulation of affinity of EGCG for luciferin

Luciferin (PubChem CID 92934) and EGCG (PubChem CID 65064) structure models were downloaded from PubChem. The protein backbone was retrieved from the protein database ID 1LCI. The molecules were docked to protein by Autodock Vina with an exhaustiveness of 8. The size of the docking grid was set to $16 \times 20 \times 24$ with a spacing of 0.375 Å. The center of docking was set in accordance with the studies therein[57,58]. The docked model with the lowest simulated affinity was selected for analysis.

## In vivo, transfection using lead MPN NPs and mScarlet3

The lead mRNA-MPN NPs (20k linear PEG, selected mRNA, EGCG, and Zr$^{IV}$ at a mass ratio of 100:1:100:2.5) were prepared as per the protocol described in Section Fabrication of mRNA-MPN NPs.

Mice were sourced from the Bioresources Facility from Peter Doherty Institute (Melbourne, Australia) and Australian BioResources (New South Wales, Australia) and housed on a 12 h light/dark cycle with ad libitum access to food and water.

Briefly, mRNA-MPN NPs were injected into C57BL/6J mice (female, 8–10-weeks-old) or B6.Cg-Gt(ROSA)26Sor$^{tm14(CAG-tdTomato)Hze}$/J mice[48] (Common name: Ai14; mixed gender, 8–10-weeks-old) at a dosage of 5 μg of mRNA per mouse (~0.25 mg kg$^{-1}$) via the lateral tail vein, while the same volume of DPBS was injected as a negative control. The mice

were euthanized by $CO_2$ asphyxiation at designed time points. Major organs, i.e., liver, kidney, lung, heart, spleen, and brain, were harvested post-injection, followed by IVIS imaging (PerkinElmer, USA). Spectral unmixing was conducted using DPBS-treated mice to unmix the auto-tissue fluorescence. Five biologically independent mice were included in each group.

### Biodistribution in mice using lead mRNA-MPN NPs
For the NP biodistribution studies, EGCG was premixed with a fluorescence dye (Rh800; excitation/emission wavelengths: 682/704 nm; EGCG:Rh800 = 5:1 (w/w)) during particle synthesis to enable in vivo tracking of the mRNA-MPN NPs according to the fluorescence signal. mRNA-MPN NPs were injected at a dosage of 5 μg of mRNA per mouse via the lateral tail vein. The mice were euthanized by $CO_2$ asphyxiation 4, 24, or 48 h post injection and the major organs were collected. Three biologically independent mice were included in each group.

### In vivo transfection and biodistribution of mRNA-SM-102 LNPs
mRNA-SM-102 LNPs were synthesized using a NanoAssemblr Ignite (Precision NanoSystems, Canada). The ethanol phase contained SM-102, DSPC, cholesterol, DMG-PEG-2000, and DiR at a molar ratio of 50:10:38.5:1.5:0.2, while mRNA was suspended in 10 mM citrate buffer (pH 4) to form the aqueous phase. The two phases were combined via microfluidics mixing at a nitrogen-to-phosphate ratio of 6:1 and an aqueous-to-ethanol ratio of 3:1 at a flow rate of 12 mL min⁻¹. Post assembly, the LNPs were dialyzed overnight in DPBS using a 6 kDa cutoff dialysis tube (Sigma-Aldrich, USA) and were stored at 4 °C. The size of the LNPs was characterized as 44 ± 4 nm using Zetasizer Nano-ZS, and mRNA encapsulation efficiency was determined to be 93 ± 2% using the Quant-iT RiboGreen RNA assay and the Infinite M200 microplate reader (excitation/emission wavelengths:480/520 nm). In vivo, transfection and biodistribution of the mRNA-SM-102 LNPs were investigated under the same conditions used for the mRNA-MPN NPs for comparative analysis.

### Elemental analysis (ICP-MS)
mRNA-MPN NPs were injected into C57BL/6J mice (mixed gender, 8–10-weeks-old) at a dosage of 0.25 mg kg⁻¹ via the lateral tail vein on day 0. For single injection groups, the treated mice were sacrificed after 4 h, 1 day, 3 days, 6 days, and 10 days post-injection. For double injection groups, a second injection with the same dose was performed on day 5, and the mice were sacrificed on day 6 and day 10. Urine samples were collected before euthanasia. The mice were euthanized using a slow influx of $CO_2$, after which cervical dislocation was carried out. After sacrificing the animals, blood was collected via cardiac puncture. Six organs, including the liver, brain, kidney, lung, spleen, and heart, were resected surgically.

The fresh organs and body fluids were then weighed and digested in 70% $HNO_3$ (0.6 mL) at 90 °C for 30 min, then diluted to achieve a $HNO_3$ concentration of 2%. Caution! Extreme care should be taken when handling 70% $HNO_3$, which can only be used in a fume hood. Zr standards were prepared using ICP multi-element standard solution IV (Merck, Germany). Zr content in the digested samples and the standard was quantified on an ICP-MS Vitesse instrument (Nu Instruments, UK), which was tuned and optimized for the detection of isotopes around 100 amu with standard tuning solution (Nu Instruments, UK), and on a NexION 2000 ICP-MS instrument (PerkinElmer, USA). Zr amount in organs and body fluids was calculated as the mass of Zr as a fraction of the injected dose and normalized for organ weight (%ID g⁻¹).

### Effect of NP building block on in vivo mRNA transfection
mRNA-MPN NPs were engineered by varying the Zr$^{IV}$-to-EGCG mass ratio, type of metal ions, and type of PEG, while mScarlet3 was employed to evaluate the transfection efficiency. All prepared mRNA-MPN NPs were injected at a dosage of 5 μg of mRNA per mouse via the lateral tail vein.

### In vivo transfection and sectioned organ analysis
For tissue section imaging, a far-red protein-encoding mRNA (emiRFP670; excitation/emission wavelengths: 642/670 nm) was loaded in mRNA-MPN NPs to reduce the interference of tissue auto-fluorescence. To further enhance the overall protein expression, two injections of mRNA-MPN NPs with a dosage of 5 μg mRNA per injection were administered to C57BL/6J mice (female, 8–10-weeks-old) intravenously at 0 h and 12 h. Thirty-six hours after the first injection, the mice were euthanized by $CO_2$ asphyxiation, and the major organs were collected. Three biologically independent mice were included in each group.

The harvested organs were fixed with 4% PFA (4 °C, overnight), dehydrated with 15% and 30% sucrose solution until the organs settled down, followed by embedding in a frozen optimal cutting temperature medium (Tissue-Tek, #4583). The entire tissue block was frozen in dry ice and cut into sections using a Leica Biosystems CM1950 Cryostat (Leica Biosystems, AU) with a thickness of 10 μm. The sections were placed on glass slides for subsequent H&E staining and confocal microscopy analysis. For cytokine assays, plasma was collected 24 h post-treatment by cardiac puncture followed by centrifugation (900×g, 15 min). The levels of IL-6 and TNF-α were quantified using enzyme-linked immunosorbent assay kits (Thermo Fisher Scientific, #88-7064-22 and #BMS607-3).

Note: Changes in size, ζ-potential, and ratio among NP constituents in response to alterations in the NP composition were recorded and summarized in Supplementary Fig. 37 and Supplementary Tables 1 and 3–6.

### Statistics and reproducibility
Statistical analyses were performed using GraphPad Prism V.9 and V.10 (GraphPad Software). Statistical analysis details are provided in the caption of each figure. Unpaired t-tests were applied to compare two datasets. One-way and two-way ANOVA tests were used for more than two datasets, unless stated otherwise. Statistical analysis was performed with a 95% confidence interval. Data are expressed as mean values ± SD. ****$p < 0.0001$; ***$p < 0.001$; **$p < 0.01$; *$p < 0.05$; ns, $p > 0.05$. The results were replicated independently at least three times. No sample-size calculations were performed. Sample size was determined to be adequate based on the magnitude and consistency of measurable differences between groups and no data were excluded from the analyses. Mice were randomized into different groups before being assayed. The investigators remained unblinded during both experiments and outcome assessment.

### MIRIBEL
The studies conducted herein, including material characterization, biological characterization, and experimental details, conform to the MIRIBEL reporting standard for bio-nano research, and we include a companion checklist of these components in the Supplementary Information.

### Reporting summary
Further information on research design is available in the Nature Portfolio Reporting Summary linked to this article.

## Data availability
All relevant data supporting the findings of this study are available within the paper and Supplementary Information. Source data are provided in this paper.

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

## Acknowledgements

F.C. acknowledges the award of a National Health and Medical Research Council Leadership Fellowship (GNT2016732). J.C. acknowledges the award of a 2024 Materials Characterization and Fabrication Platform (MCFP) Sustainability Research Grant for the training and use of NexION 2000 and Vitesse. This research was partially funded by the Victoria State Government through mRNA Victoria (BMIN-2-22-21953; F.C.) and the Cumming Global Centre for Pandemic Therapeutics (CGCPT00046; F.C., C.C.-J., and J.C.). This work was performed in part at the MCFP, the Victorian Node of the Australian National Fabrication Facility (ANFF), and the Peter Doherty Institute for Infection and Immunity. TEM analyses and EDX experiments were conducted at the Bio21 Advanced Microscopy Facility at the University of Melbourne. H&E staining was conducted at the Melbourne Histology Platform, The University of Melbourne. We thank Denzil Furtado, Paul Brannon, Dr. Wanjun Xu, Dr. Shiyao Li, Dr. Mai Ngoc Vu, and Anthony Ngadiyoto for their helpful discussions.

## Author contributions

Y.G. and J.C. contributed equally to this work. J.C. and F.C. conceived the project. Y.G. and J.C. designed, executed the experiments, and analyzed the data. Z.W. provided in vivo experimental instructions and was involved in the experiments. C.L. synthesized the LNPs and performed the simulation of molecule affinity. T.W. performed TEM and EDX mapping. C.-J.K. performed AFM experiments and analysis. H.D. was involved in the in vitro experiments. D.N.J. provided experimental instructions for elemental analysis. S.F. and R.D.R. provided instructions for the mice experiments and data analysis. C.C.-J and F.C. directed the project. Y.G., J.C., C.C.-J., and F.C. wrote the manuscript. All authors reviewed the manuscript.

## Competing interests

F.C., J.C., Y.G., and C.C.-J. have filed a patent application for this technology. F.C. is a shareholder of Messenger Bio Pty Ltd. The remaining authors declare no competing interests.
