## [Transparent Peer Review file · Nature Communications]

mRNA Delivery Enabled by Metal–Organic Nanoparticles

Corresponding Author: Professor Frank Caruso

Version 0:

Reviewer comments:

Reviewer #1

(Remarks to the Author)

The article titled " mRNA Delivery Enabled by Metal–Organic Nanoparticles" presents an interesting strategy for the delivery of mRNA with mRNA-MPN NPs contained PEG, metal ions and phenolic compounds. This strategy will be considered as a followed-up work of their previous work of bioactive metal–phenolic NPs (b-MPN NPs) and use mRNA as the example of biomacromolecules. In this work, the author included more in vitro and in vivo data to support their application in mRNA delivery that can provide better mRNA transfection efficiency. However, the novelty of this delivery platform and the effects of metal ions in the delivery system are a concern for this work. Moreover, the metal ions seem to have negative effects on the mRNA transfection therefore the authors need to consider whether this platform suitable for mRNA delivery. Thus, a major revision is recommended.

1. The novelty of this work. Similar work of PEG-MPN NPs¹ with "Bioactive macromolecules, including small-interfering RNA (siRNA) and single or multiple proteins with distinct isoelectric points (ranging from 4.2 to 10.3) and molecular weights (Mw, 12.4–240 kDa)" has been reported by the same group. The author needs to justify the novelty of their mRNA-MPN NPs with bioactive metal–phenolic NPs (b-MPN NPs).

2. The role of PEG. The author claims the PEG is "a seeding agent that can increase the local concentrations of precursors (mRNA and polyphenols) and drive the formation of NPs under ambient conditions". But MPN NPs can also work as delivery system for "potential biomedical applications" without seeding agents². Can the authors provide any reasons for introducing PEG in the delivery system rather than only MPN NPs? Or provide the comparison of these two platforms to support why PEG embedded mRNA-MPN NPs is better than MPN NPs?

3. In Figure 1g, further data points are necessary to validate the impact of metal ions on mRNA transfection efficiency. The pattern suggests an initial increase followed by a decrease in influence. Initially, with no added metal ions, the efficiency stands at 78%. Upon the introduction of metal ions, particularly at a ratio of 1:40, efficiency spikes to 94.5%. However, subsequent to this peak, efficiency begins to decline. Notably, there is a gap between the 0:40 and 1:40 ratios, leaving the trend unsubstantiated. Moreover, a single measurement lacks the robustness required to firmly establish this pattern. Therefore, the author should conduct a minimum of three repetitions for each condition to ensure the accuracy and rigor of the observed trends.

4. In Figure 1i, to comprehensively evaluate the impact of various metal ions on mRNA transfection efficiency, it is imperative for the author to incorporate a control group lacking any metal ions. This control serves as a baseline reference, enabling the determination of whether the presence of metal ions exerts a positive or negative influence on transfection efficiency. Without this essential comparison, it becomes challenging to accurately interpret the effects observed with the different metal ion treatments.

5. Figure 1h metal ions do seem to have a negative impact on mRNA transfection. Have the authors considered removing metal ions from this system? Or the author can provide any evidence that the introducing of metal ion can actually improve the transfection.

6. In animal experiments, it's essential for the author to explore the long-term accumulation of metals to determine whether they persist or are metabolized over time. By extending the metal accumulation period, the author can assess whether metals continue to accumulate or if there's a plateau or decline due to metabolic processes. Additionally, investigating the distribution of metals in urine and blood provides valuable insights into systemic clearance mechanisms and potential routes

of elimination.

Examining the impact of increasing the number of doses on metal concentration is crucial for understanding dose-dependent effects. This investigation can shed light on whether repeated exposure leads to cumulative metal accumulation or if the body's clearance mechanisms effectively regulate metal levels over time.

Reference

1. Chen, J.; Pan, S.; Zhou, J.; Lin, Z.; Qu, Y.; Glab, A.; Han, Y.; Richardson, J. J.; Caruso, F., Assembly of bioactive nanoparticles via metal–phenolic complexation. *Adv. Mater.* 2022, 34 (10), 2108624.
2. Xu, W.; Lin, Z.; Pan, S.; Chen, J.; Wang, T.; Cortez-Jugo, C.; Caruso, F., Direct Assembly of Metal-Phenolic Network Nanoparticles for Biomedical Applications. *Angew. Chem.* 2023, 135 (45), e202312925.

Reviewer #2

(Remarks to the Author)

In this manuscript, Gu et al, report versatile, noncationic metal-organic-based nanoparticles for mRNA delivery. By screening a range of components and relative compositional ratio, they identified a lead formulation for efficient mRNA transfection both in vitro and in mice. The organ tropism can be tuned by varying nanoparticle composition, and intravenous injection of the newly developed metal-organic nanoparticles achieved predominant mRNA expression and gene editing across the liver, kidney, and brain. The study is interesting, and the manuscript is written concisely, but several critical issues should be addressed before publication.

1. Agarose gel electrophoresis was performed to characterize the mRNA loading efficiency of nanoparticles, which is not a quantitative measurement. I suggest the authors use RiboGreen to quantify the encapsulated mRNA in nanoparticles.
2. The SM-102 LNP formulation used in this study is not its original formulation (the molar ratio of SM-102: Cholesterol: DSPC: DMG-PEG should be 50:38.5:10:1.5). What is the impact of LNP formulation on mRNA delivery? Can the authors compare it with the original SM-102 formulation? SM-102 is a liver-targeting LNP, while in Fig. 4d, mRNA expression was observed in almost all organs. I suggest the authors use firefly luciferase-encoding mRNA to do the in vivo mRNA transfection and biodistribution experiments to avoid the interference of tissue auto-fluorescence.
3. Fig. 2e, in vitro mRNA release experiment, how is the structural integrity of the released mRNA?
4. Fig. 6f-h, the authors highlight that the organ tropism of the NPs is influenced by the type of metal ions, can the authors comment on the potential mechanism? I wondered whether the incorporation of different metal ions may alter the physical-chemical properties of the NPs.

Reviewer #3

(Remarks to the Author)

In their manuscript, the Caruso et. al. report a versatile, non-cationic nanoparticle platform consisting of a poly(ethylene-glycol)-polyphenolic network that can package and deliver mRNA payloads. In brief, the authors develop a library of metal-organic nanoparticles with robust mRNA transfection in vitro and in mice (mRNA-MPN NPs). Key studies in the manuscript detail the assembly of their mRNA-MPN NP platform using formulation screening and optimization, characterization of the physicochemical properties of their mRNA-MPN NPs, in vitro evaluation of their mRNA-MPN NP platform, in vivo evaluation of their mRNA-MPN NP platform (including expression studies relative to Moderna's SM-102 LNP, biocompatibility studies, and further in vivo performance studies detailing mRNA expression using their highly performing formulations). Overall, the work by Caruso et. al. is innovative, interesting, and thorough. Their system is both well-characterized and affords interesting mRNA delivery properties that are distinct from the SM-102 formulations. Notably, their mRNA-MPN NP platform also delivers mRNA payloads to the brain following IV injections, opening novel therapeutic avenues for mRNA therapies. Further, the manuscript is well-written and well-organized, and it is the opinion of this reviewer that this manuscript will be of interest to the broader mRNA delivery community. Based on this, this reviewer recommends publication with minor revisions following consideration of the following points:

- 1) Page 5, Line 171 - the authors note that: "...increasing the concentration of metal ions beyond a mass ratio of 1:40 inversely impacted the transfection efficiency (73% and 4% at ZrIV-to-EGCG mass ratios of 3:40 and 10:40, respectively) despite the higher mRNA loading observed at higher ZrIV concentrations." Could the authors provide further commentary on this observation and speculate as to why this may be the case?
- 2) Figure 4c – it is a bit difficult for this reviewer to understand the key take away messages from this figure. Are there additional ways that this data may be represented (even if in the supporting information) that may improve comprehension for some readers?
- 3) Figure 4d – the delivery profile for mRNA-MPN NPs to the brain is exciting. One question from this reviewer is about the background signal observed in the brain from DPBS. Could the authors provide additional brief commentary on where this background signal may arise from?
- 4) Figure 4d – once again, the delivery profile of mRNA-MPN NPs to the brain is exciting. Could the authors provide further commentary speculating on how/why their mRNA-MPN NPs are better able to deliver mRNA to the brain than the SM-102 formulation? Also, could the authors provide further commentary as to what potential therapeutic avenues these results may

open?

Version 1:

Reviewer comments:

Reviewer #1

(Remarks to the Author)

The authors have answered all the questions, and I agreed to this manuscript to be published in Nature communications.

Reviewer #2

(Remarks to the Author)

The authors have addressed most of my concerns with additional data and more discussion in their revised manuscript. I recommend the publication of the revised manuscript in Nature Communications.

Reviewer #3

(Remarks to the Author)

The authors have sufficiently addressed my questions

REVIEWER #1

The article titled "mRNA Delivery Enabled by Metal–Organic Nanoparticles" presents an interesting strategy for the delivery of mRNA with mRNA-MPN NPs contained PEG, metal ions and phenolic compounds. This strategy will be considered as a followed-up work of their previous work of bioactive metal–phenolic NPs (b-MPN NPs) and use mRNA as the example of biomacromolecules.

In this work, the author included more *in vitro* and *in vivo* data to support their application in mRNA delivery that can provide better mRNA transfection efficiency. However, the novelty of this delivery platform and the effects of metal ions in the delivery system are a concern for this work. Moreover, the metal ions seem to have negative effects on the mRNA transfection therefore the authors need to consider whether this platform suitable for mRNA delivery. Thus, a major revision is recommended.

1. The novelty of this work. Similar work of PEG-MPN NPs with “Bioactive macromolecules, including small-interfering RNA (siRNA) and single or multiple proteins with distinct isoelectric points (ranging from 4.2 to 10.3) and molecular weights (Mw, 12.4–240 kDa)” has been reported by the same group. The author needs to justify the novelty of their mRNA-MPN NPs with bioactive metal–phenolic NPs (b-MPN NPs).

Response: We appreciate the reviewer’s comments on the novelty of our work. Our work aims to address two major challenges faced by current mRNA delivery platforms: i) the use of non-cationic moieties to complex mRNA, unlike cationic lipids used in mRNA vaccines, which have concerns of toxicity/inflammation; and ii) limited organ tropism (most nanoparticles are trapped in the liver and/or spleen post systemic administration). Our prior work on metal–organic nanoparticles was on protein and siRNA delivery *in vitro* (*Adv. Mater.* **2022**, *34*, 2108624), whereas in the current work, we report i) the development of a *highly biocompatible and noncationic* nanoparticle platform that can readily assemble mRNA and enable robust transfection both *in vitro* and in mice; and ii) *diverse organ tropism* that can be modulated by varying nanoparticle compositions. As a biomolecule, mRNA is significantly different from proteins and siRNA; ensuring sufficient loading and functionality of mRNA required precise nanoengineering of the metal–organic nanoparticles. In the current work, comprehensive formulation screening was also performed to realize their efficacy *in vivo*.

We described the abovementioned novelty and significance in the INTRODUCTION and CONCLUSIONS sections of the original manuscript as follows:

INTRODUCTION: “[...] *Notably, the organ tropism of protein expression (i.e., organs that exhibit predominant protein expression) can be adjusted simply by altering the composition and ratio of the NP building blocks. The noncationic nature, high biocompatibility, mRNA delivery efficacy, and modularity of the mRNA-MPN NPs not only provide a promising alternative to current mRNA delivery platforms but also open avenues for the development of future NP-enabled therapeutics.*”

CONCLUSIONS: “[...] *Therefore, we envision that the NP platform reported herein will further advance NP-mediated therapeutics, widening the breadth of diseases that can be addressed with mRNA NP technologies.*”

To further clarify the novelty, we have now updated the Abstract of the revised manuscript as follows:

“However, current mRNA delivery platforms face some challenges, including limited organ tropism for nonvaccine applications and inflammation induced by cationic nanoparticle components. Herein, we address these challenges through a versatile, noncationic nanoparticle platform, whereby mRNA is assembled into a poly(ethylene glycol)-polyphenol network stabilized by metal ions.”

“Intravenous administration of the lead mRNA-containing metal–organic nanoparticles enables [...], and kidney, while organ tropism is tuned by varying nanoparticle composition.”

2. The role of PEG. The author claims the PEG is “a seeding agent that can increase the local concentrations of precursors (mRNA and polyphenols) and drive the formation of NPs under ambient conditions”. But MPN NPs can also work as delivery system for “potential biomedical applications” without seeding agents. Can the authors provide any reasons for introducing PEG in the delivery system rather than only MPN NPs? Or provide the comparison of these two platforms to support why PEG embedded mRNA-MPN NPs is better than MPN NPs?

Response: We thank the reviewer for raising this point. We incorporated PEG for the assembly of mRNA-MPN NPs in this work for three main reasons:

(i) Improved cargo stability during systemic circulation. The PEG-free MPN NPs reported in *Angew. Chem. Int. Ed.* **2023**, 62, e202312925 and the bioactive MPN NPs (b-MPN NPs; with PEG) reported in *Adv. Mater.* **2022**, 34, 2108624 have distinct assembly and cargo incorporation strategies. For the PEG-free MPN NPs, the cargo is post-adsorbed onto the NP surface after NP formation. Hence, a fragile molecule, such as mRNA, would be prone to dissociation (from MPN NPs) and degradation during systemic circulation. In contrast, the assembly of b-MPN NPs (with PEG) allows the encapsulation of cargo, including mRNA, during NP formation. Therefore, the cargo is more likely to be encapsulated within b-MPN NPs, affording higher stability.

We have now updated a sentence regarding this point on page 2 of the revised manuscript as follows:

“[...] (3) a seeding agent [...]. The engineered mRNA-MPN NPs stabilize mRNA within a metal–organic network and display superior [...]”

(ii) Increased yield of mRNA-MPN NPs. Compared to mRNA-MPN NPs without PEG, the NPs that include PEG displayed increased NP yield from 2×10^8 particles mL⁻¹ to 4×10^9 particles mL⁻¹.

We have now added this result as Supplementary Fig. 1b (new figure part) in the revised Supplementary Information (SI). The associated figure caption has been updated as follows:

“Supplementary Fig. 1| **a**, [...] suggesting complexation between Zr^{IV} and EGCG. **b**, Yield of mRNA-MPN NPs prepared with or without the inclusion of PEG. The turbidity of the NP suspension provided a qualitative indication of NP concentration while the number of mRNA-MPN NPs was quantified by NanoSight N300 (Malvern Panalytical, UK).”

We have also updated the associated discussion on page 3 of the revised manuscript as follows:

“PEG was employed as a seeding agent for mRNA-MPN NP assembly to increase the local concentration of precursors²⁹, yielding 4×10^9 mRNA-MPN NPs per milliliter (Supplementary Fig. 1b). Composition screening showed [...]”

(iii) Greater chemical diversity for tuning mRNA expression tropism. Including PEG as the building block allows greater chemical diversity of mRNA-MPN NPs, which can be leveraged to further adjust mRNA expression tropism. In Fig. 6i and on page 16 (lines 474–478) of the original manuscript (page 18 of the revised manuscript), we demonstrated that using 2k 4-arm PEG increased splenic expression when compared with 20k linear PEG as follows:

“[...] Using 2k 4-arm PEG also resulted in an overall diminished level of mRNA expression compared to using 20k linear PEG (Supplementary Fig. 32), but the splenic expression increased from 10% to 50% with the normalized radiance efficiency increasing from 4-fold to 16-fold (normalized to DPBS-treated mice) (Fig. 6i and Supplementary Fig. 32). [...]”

No change was made regarding this point.

3. In Figure 1g, further data points are necessary to validate the impact of metal ions on mRNA transfection efficiency. The pattern suggests an initial increase followed by a decrease in influence. Initially, with no added metal ions, the efficiency stands at 78%. Upon the introduction of metal ions, particularly at a ratio of 1:40, efficiency spikes to 94.5%. However, subsequent to this peak, efficiency begins to decline. Notably, there is a gap between the 0:40 and 1:40 ratios, leaving the trend unsubstantiated. Moreover, a single measurement lacks the robustness required to firmly establish this pattern. Therefore, the author should conduct a minimum of three repetitions for each condition to ensure the accuracy and rigor of the observed trends.

Response: We thank the reviewer for the suggestion. In response to the gap between the existing Zr^{IV}-to-EGCG mass ratios (0:40, 1:40, 3:40, 10:40), we have now evaluated the mRNA transfection efficacy at Zr^{IV}-to-EGCG ratios of 0.3:40, 0.5:40, 0.8:40, 2:40, and 5:40 to further clarify the trend on the impact of metal ions. Increasing the proportion of Zr^{IV} gradually increased the transfection efficiency and mean fluorescence intensity (MFI), with a peak observed at the mass ratio of 1:40. However, further increasing Zr^{IV} beyond this ratio led to a gradual reduction in both transfection efficiency and MFI. To explore the possible reasons, we evaluated the mRNA loading and release profiles corresponding to these ratios. The loading increased with higher Zr^{IV}-to-EGCG mass ratios and plateaued at 1:40, while the release profile displayed a similar trend to that observed in mRNA transfection. This suggests that a balanced loading and release at the 1:40 ratio is optimal for mRNA transfection.

We have now updated Fig. 1g, h in the revised manuscript, Supplementary Fig. 3e, f, and Supplementary Figure 7 (updated a; newly added b–d) in the revised SI. The associated discussion has now been updated on pages 5–6 of the revised manuscript as follows:

“We next investigated the effect of the metal ion-to-phenolic ligand (EGCG) mass ratio [...] mass ratio of 100. Compared to the NPs with no metal ion, the inclusion of Zr^{IV} with Zr^{IV}-to-EGCG mass ratios up to 1:40 gradually increased the transfection efficiency from 78 to 95%, and the MFI by 2.5-fold (Fig. 1g, h, and Supplementary Fig. 7a–c). The integration of metal ions also increased the mRNA loading (Fig. 1h inset and Supplementary Fig. 3e, f). However, increasing the concentration of metal ions beyond a mass ratio of 1:40 inversely impacted the transfection efficiency (reducing to 4% at a Zr^{IV}-to-EGCG mass ratio of 10:40) despite the higher mRNA loading observed at higher Zr^{IV} concentrations (Fig. 1h inset and Supplementary

Fig. 3e, f). To investigate this further, mRNA release corresponding to different Zr^{IV}-to-EGCG mass ratios was evaluated using fluorescently labelled mRNA. As shown in Supplementary Fig. 7b–d, mRNA release decreased at Zr^{IV} concentrations beyond mass ratios of 1:40 (i.e., ~0.75 μg mRNA was released at 1:40, whereas only ~0.55 and ~0.3 μg mRNA was released at 2:40 and 10:40, respectively). This decrease is consistent with the reduction observed in *in vitro* mRNA transfection at the same ratios. This suggests that higher amounts of Zr^{IV} within mRNA-MPN NPs (i.e., Zr^{IV}-to-EGCG mass ratio >1:40) may increase the cross-linking density of mRNA with NPs, thereby hindering mRNA release and reducing the efficacy of mRNA transfection. In addition, a higher metal concentration is likely to induce fluorescence quenching (Supplementary Fig. 8), thus further reducing the MFI of transfected cells. Collectively, the Zr^{IV}-to-EGCG mass ratio of 1:40 might exert balanced influences on mRNA loading, release, and fluorescence quenching, making it optimal for mRNA transfection.

We have also updated the ACKNOWLEDGEMENT section regarding this analysis as follows:

“We thank Denzil Furtado, Paul Brannon, Dr. Wanjun Xu, Dr. Shiyao Li, and Anthony Ngadiyoto for helpful discussions.”

Regarding the comment on the number of measurements, the number of repetitions in the caption of Fig. 1 of the original (and revised) manuscript was stated as follows:

“*All experiments were performed in triplicates (n = 3), and data are presented as mean ± standard deviation (SD).*”

Therefore, no change was made regarding this comment.

4. In Figure 1i, to comprehensively evaluate the impact of various metal ions on mRNA transfection efficiency, it is imperative for the author to incorporate a control group lacking any metal ions. This control serves as a baseline reference, enabling the determination of whether the presence of metal ions exerts a positive or negative influence on transfection efficiency. Without this essential comparison, it becomes challenging to accurately interpret the effects observed with the different metal ion treatments.

Response: We thank the reviewer for the suggestion. The mRNA-MPN NP platform reported herein exhibits a high degree of modularity in terms of a rich choice of building blocks. Varying the NP composition (i.e., with or without metal ions, metal ion-to-phenolic ligand mass ratio, type of metal ions) allowed for alterations in mRNA expression tropism in mice. For example, mRNA-MPN NPs without metal ions, as one of the studied formulations, led to negligible mScarlet3 expression in most harvested organs, suggesting the positive influence of metal ions in improving the overall level of mRNA transfection. This result was included in Fig. 6c and discussed on page 15 (lines 438–442) of the original manuscript (page 17 of the revised manuscript) as follows:

“*[...] For example, using mRNA-MPN NPs without metal ions (i.e., Zr^{IV}-to-EGCG mass ratio = 0:40) led to a negligible mScarlet3 fluorescence signal or expression in most harvested organs, suggesting the vital role of metal ions in improving the overall level of mRNA transfection. However, this ratio resulted in a higher mScarlet3 expression in the brain [...]*”

Furthermore, the objective of the *in vitro* screening was to identify potential formulations that enable mRNA transfection, with the most effective formulation being selected as the lead formulation. Therefore, similar to other work with mRNA delivery systems, untreated cells were

selected as a more suitable negative control and the baseline, which were included in all *in vitro* results in the original and revised manuscript (Figs. 1 and 3).

No change was made regarding this comment.

5. Figure 1h metal ions do seem to have a negative impact on mRNA transfection. Have the authors considered removing metal ions from this system? Or the author can provide any evidence that the introducing of metal ion can actually improve the transfection.

Response: We thank the reviewer for raising this point. Regarding the influence of metal ions on mRNA transfection *in vitro* (i.e., Fig. 1g, h), we refer to our response to Comment 3 (of Reviewer #1) which raised the same point. Therein, we provided evidence that introducing metal ions within the range of metal ion-to-polyphenol mass ratio from 0:40 to 1:40 positively influences mRNA transfection.

In addition to the explanation based on mRNA loading and release profiles (i.e., Response to Comment 3 of Reviewer #1), we elaborated further based on endosomal escape capability. Previous studies in our group have shown that metal-phenolic networks have a stronger buffering capacity than phenolic ligands, thus enabling a higher degree of endosomal escape of nanoparticles (*ACS Nano* **2019**, *13*, 11653; *Chem. Mater.* **2021**, *33*, 2557). Therefore, we hypothesized that incorporating metal ions may enhance the endosomal escape capability of mRNA-MPN NPs, further improving mRNA transfection efficacy.

We have now compared the colocalization of metal-containing and metal-free mRNA-MPN NPs with endo/lysosomes *via* super-resolution microscopy and Pearson's correlation coefficient analysis. The results showed that metal-containing mRNA-MPN NPs (i.e., lead formulation) showed a lower degree of colocalization with endo/lysosomes than the metal-free mRNA-MPN NPs, indicating greater endosomal escape. We have now updated Supplementary Fig. 17 and added a new Figure (Supplementary Fig. 18) in the revised SI to include these results. A discussion regarding this point is also provided in the endosomal escape-relevant section on page 9 of the revised manuscript as follows:

“[...], indicating endosomal escape of mRNA. The endosomal escape ability of mRNA-MPN NPs is likely attributed to the buffering capacity imparted by MPNs. Although incorporating polyphenols into NPs can facilitate endosomal escape of mRNA^{39,40} (Supplementary Fig. 18), metal-phenolic coordination exerts synergistic buffering effects, enabling superior endosomal escape capability *via* enhanced proton-sponge effects^{41,42} (Supplementary Figs. 17 and 18).”

Furthermore, including metal ions in mRNA-MPN NPs improves the overall level of mRNA expression *in vivo*, as described on page 15 (lines 438–441) of the original manuscript (page 17 of the revised manuscript) as follows:

“For example, using mRNA-MPN NPs without metal ions (i.e., Zr^{IV}-to-EGCG mass ratio = 0:40) led to a negligible mScarlet3 fluorescence signal or expression in most harvested organs, suggesting the vital role of metal ions in improving the overall level of mRNA transfection.”

More importantly, mRNA-MPN NPs is a four-component NP platform where metal ions play an essential role in the chemical diversity and programming the mRNA expression tropism in mice. In Fig. 6f–h and page 16 (lines 463–470) of the original manuscript (page 18 of the revised manuscript), we described the modulation of mRNA expression tropism in the liver, kidney, and lung by varying metal ions as follows:

“The type of metal ions influenced the organ tropism of expressed mRNA (Fig. 6f–h). Replacing Zr^{IV} with Co^{II} resulted in <5% mRNA expression in the liver and kidney but ~50% expression in the lung (i.e., ~50%, 13-fold higher radiance efficiency than DPBS-treated mice). In contrast, [...] and lung (i.e., ~11%, 4.5-fold higher radiance efficiency than DPBS-treated mice).”

Collectively our findings show that the introduction of metal ions (at suitable ratios) overall positively influences mRNA transfection and allows tuning of mRNA expression tropism.

6. In animal experiments, it's essential for the author to explore the long-term accumulation of metals to determine whether they persist or are metabolized over time. By extending the metal accumulation period, the author can assess whether metals continue to accumulate or if there's a plateau or decline due to metabolic processes. Additionally, investigating the distribution of metals in urine and blood provides valuable insights into systemic clearance mechanisms and potential routes of elimination.

Examining the impact of increasing the number of doses on metal concentration is crucial for understanding dose-dependent effects. This investigation can shed light on whether repeated exposure leads to cumulative metal accumulation or if the body's clearance mechanisms effectively regulate metal levels over time.

Response: We appreciate the reviewer's suggestions. We have now extended the metal accumulation study to 10 days, which is the maximum treatment period approved by our Animal Ethics 27608. Approximately 0.8 mg kg^{-1} of Zr was introduced for every 0.25 mg kg^{-1} of mRNA delivered to mice, resulting in only nanogram-level changes in Zr content per gram of organ or body fluid under experimental conditions. The Zr content in the blood peaked at 4 h and gradually reduced thereafter, suggesting the presence of Zr in circulation within 4 h post injection. Meanwhile, the rapid decrease of Zr in the urine indicates efficient metal excretion from mice. For other major harvested organs, although Zr accumulation peaked on either day 3 (e.g., liver, kidney, spleen, heart) or day 6 (e.g., lung, brain), it reduced to the baseline level (i.e., metal content in DPBS-treated mice) on day 10, suggesting negligible Zr accumulation owing to the NPs within 10 days. Furthermore, the mice exhibited negligible weight loss across the 10-day study, further implying the safety of mRNA-MPN NPs during the given treatment period.

The dose-dependent effect on metal accumulation was examined by applying a second injection on day 5. The Zr content slightly increased in the urine, blood, and most harvested organs on day 6 and was then excreted by day 10. Overall, the percentage of the injected dose of Zr per gram tissue (%ID g^{-1}) in harvested organs and body fluids was low (i.e., <2%) regardless of single or double injections, which further implies minimal *in vivo* metal accumulation.

The graphs shown in Fig. 5c–e in the original manuscript have been replaced by Fig. 5 c, which now includes the new results. The new data are also shown as Supplementary Figs. 29 and 30 (new figures) in the revised SI. The discussion has been updated on page 15 of the revised manuscript as follows:

“Furthermore, mice exhibited negligible weight loss within 10 days post IV administration (Supplementary Fig. 29). To assess any potential toxicity of the metal ions in the NPs, the biodistribution of Zr in organs was assessed up to 10 days post administration using inductively coupled plasma mass spectrometry (ICP-MS) on digested organ and body fluid (i.e., urine and blood) preparations. [...] However, under experimental conditions, NPs resulted in only

nanogram-level changes in Zr content per gram of organ or body fluid (Fig. 5c and Supplementary Fig. 30). At 4 h post injection, approximately 60 ng of Zr was detected per gram of blood, which was the highest level among all tested time points within 10 days (Fig. 5c), suggesting the presence of Zr in circulation within 4 h. The gradual reduction of Zr content in blood after 4 h indicates gradual organ deposition or excretion. In the urine samples, ~70 ng of Zr was detected per gram of urine at 4 h, which decreased rapidly thereafter (Fig. 5c and Supplementary Fig. 30), suggesting efficient metal excretion from mice. Although Zr signals in harvested organs peaked on either day 3 (e.g., liver, kidney) or day 6 (e.g., brain, lung), they reduced to the baseline level (i.e., Zr content in DPBS-treated mice), implying that the overall Zr accumulation owing to the NPs was negligible within the 10-day study. The dose-dependent effect on Zr accumulation was also evaluated by applying a second injection on day 5. The Zr content slightly increased in the urine, blood, and most harvested organs on day 6 and was excreted by day 10 (Supplementary Fig. 30). Overall, the percentage of the injected dose of Zr per gram tissue (%ID g⁻¹) in harvested organs and body fluids was low (i.e., <2%) regardless of single or double injections, further implying minimal *in vivo* metal accumulation. Together these data highlight the safety and [...].”

The caption of the new Fig. 5c has been updated as follows:

“**Fig. 5 | *In vivo* biocompatibility of and metal excretion from mRNA-MPN NPs.** [...] c, Zr excretion profiles from harvested organs and body fluids (i.e., urine and blood) post treatment with DPBS or mRNA-MPN NPs, as measured by ICP-MS elemental analysis. Zr excretion is presented as the amount of Zr per gram of organ or body fluid collected from mice. Statistical significance was analyzed using two-way ANOVA: ns, $P > 0.05$.”

We have also updated the corresponding METHODS sections (i.e., *In vivo transfection using lead MPN NPs and mScarlet 3* and *Elemental analysis (ICP-MS)*) on pages 23 and 24 of the revised manuscript as follows:

On Page 23:

***In vivo* transfection using lead MPN NPs and mScarlet3**

“[...] (Ethics 10404 and 27608). Mice were sourced from the Bioresources Facility from Peter Doherty Institute (Melbourne, Australia) and Australian Bio Resources (New South Wales, Australia) and housed on a 12 h [...]”

On Page 24:

Elemental analysis (ICP-MS)

mRNA-MPN NPs were injected into C57BL/6J mice (mixed gender, 8–10 weeks old) at a dosage of 0.25 mg kg⁻¹ *via* the lateral tail vein on day 0. For single injection groups, the treated mice were sacrificed after 4 h, 1 day, 3 days, 6 days, and 10 days post-injection. For double injection groups, a second injection with the same dose was performed on day 5, and the mice were sacrificed on day 6 and day 10. Urine samples were collected before euthanasia. The mice were euthanized using a slow influx of CO₂, after which cervical dislocation was carried out. After sacrificing the animals, blood was collected *via* cardiac puncture. Six organs, including the liver, brain, kidney, lung, spleen and heart, were resected surgically.

The fresh organs and body fluids were then weighed and digested in 70% HNO₃ (0.6 mL) at 90 °C for 30 min, then diluted to achieve a HNO₃ concentration of 2%. *Caution! Extreme care should be taken when handling 70% HNO₃, which can only be used in a fume hood.* Zr standards were prepared using ICP multi-element standard solution IV (Merck, Germany). Zr

content in the digested samples and the standard was quantified on a ICP-MS Vitesse instrument (Nu Instruments, UK), which was tuned and optimized for detection of isotopes around 100 amu with standard tuning solution (Nu Instruments, UK), and on a NexION 2000 ICP-MS instrument (PerkinElmer, USA). Zr amount in organs and body fluids was calculated as mass of Zr as a fraction of the injected dose and normalized for organ weight (%ID g⁻¹).

We have also updated the ACKNOWLEDGEMENT section as follows:

“[...] (GNT2016732). J.C. acknowledges the award of a 2024 Materials Characterisation and Fabrication Platform (MCFP) Sustainability Research Grant for the training and use of NexION 2000 and Vitesse. This research was [...]”

REVIEWER #2

In this manuscript, Gu et al, report versatile, noncationic metal-organic-based nanoparticles for mRNA delivery. By screening a range of components and relative compositional ratio, they identified a lead formulation for efficient mRNA transfection both in vitro and in mice. The organ tropism can be tuned by varying nanoparticle composition, and intravenous injection of the newly developed metal-organic nanoparticles achieved predominant mRNA expression and gene editing across the liver, kidney, and brain. The study is interesting, and the manuscript is written concisely, but several critical issues should be addressed before publication.

1. Agarose gel electrophoresis was performed to characterize the mRNA loading efficiency of nanoparticles, which is not a quantitative measurement. I suggest the authors use RiboGreen to quantify the encapsulated mRNA in nanoparticles.

Response: We thank the reviewer for the suggestion. According to the user guide of the RiboGreen RNA Assay Kit (Thermo Fisher Scientific, R11490), salts, phenols, and PEG are regarded as common contaminations that will greatly affect assay accuracy given the components used for our NP assembly. Therefore, we did not use it for loading quantification.

To clarify this point, we have now made a note of this in the caption of Supplementary Fig. 3 in the revised SI:

“Note that the RiboGreen assay was not used for loading quantification owing to incompatibility of our particle constituents with the assay kit, as determined from the product manual (Thermo Fisher Scientific, R11490).”

In the present work, agarose gel electrophoresis was conducted ($n > 3$) primarily during the formulation screening stage to observe trends in mRNA loading. This method helped determine the optimal type of phenolic ligand and PEG, and the concentration of metal ions (Supplementary Fig. 3 in both original and revised SI). ImageJ software was used to provide semi-quantitative analysis based on the intensity, and the corresponding mRNA loading efficiency (%) was presented in Supplementary Fig. 3 of the original and revised SI. Furthermore, we also Cy5-labeled mRNA to quantify the mRNA loading efficiency based on fluorescence as a secondary method, and the results were consistent with the agarose gel electrophoresis data. These results are shown in Fig. 2d and Supplementary Fig. 13 of the original (and revised) manuscript and SI, respectively, with reference on page 6 of the original (and revised) manuscript as follows:

“To elucidate the assembly [...] Fluorescence spectroscopy showed ~50% loading efficiency, confirming the observations on agarose gels (Fig. 2d and Supplementary Fig. 13).”

2. The SM-102 LNP formulation used in this study is not its original formulation (the molar ratio of SM-102: Cholesterol: DSPC: DMG-PEG should be 50:38.5:10:1.5). What is the impact of LNP formulation on mRNA delivery? Can the authors compare it with the original SM-102 formulation? SM-102 is a liver-targeting LNP, while in Fig. 4d, mRNA expression was observed in almost all organs. I suggest the authors use firefly luciferase-encoding mRNA to do the in vivo mRNA transfection and biodistribution experiments to avoid the interference of tissue auto-fluorescence.

Response: We thank the reviewer for pointing this out. Regarding the comments on the influence of LNP formulation on mRNA delivery, we have now compared the performance of the modified SM-102 formulation that we used (i.e., SM-102:cholesterol:DSPC:DMG-PEG =

48:40:10:2) and the “original Moderna formulation” (i.e., SM-102:cholesterol:DSPC:DMG-PEG = 50:38.5:10:1.5) on *in vivo* distribution and mScarlet3 expression. As shown in **Figure 1** and **Figure 2** in the Response Letter (**for review only, not for publication**), there was no significant difference in organ tropism of LNP distribution (i.e., both formulations primarily accumulated in the liver) and mRNA expression (i.e., both formulations demonstrated predominant expression in the liver and spleen), indicating that such difference in formulation has a negligible effect on mRNA delivery.

Regarding the broad mRNA expression observed across all harvested organs for SM-102 LNPs, as shown in **Figure 2** in the Response Letter, mScarlet3 expression was observed in all harvested organs, with the liver displaying the most prominent level of mScarlet3 expression, thus consistent with the organ tropism of SM-102 LNPs. Importantly, there was no observable mScarlet3 expression in the brain with both SM-102 LNP formulations.

We have now updated all SM-102 LNP relevant figures (Fig. 4c, d in the original (and revised) manuscript and Supplementary Fig. 25 in the original SI (now Supplementary Fig. 26 in the revised SI)) using results obtained from the original Moderna formulation and further clarified the associated discussion on page 12 of the revised manuscript as follows:

“As the liver exhibited the most prominent level of mRNA expression for mRNA-MPN NPs, the *in vivo* transfection efficacy of the mRNA-MPN NPs was compared against LNPs formulated with SM-102 ionizable lipid (used in Moderna’s SARS-CoV-2 vaccine vector, which show predominant liver expression).”

We have also updated the associated METHODS section (i.e., *In vivo transfection and biodistribution of mRNA-SM-102 LNPs*) on page 23 of the revised manuscript as follows:

“[...] The ethanol phase contained SM-102, DSPC, cholesterol, DMG-PEG-2000, and DiR at a molar ratio of 50:10:38.5:1.5:0.2, while mRNA was suspended in 10 mM citrate buffer (pH 4) to form the aqueous phase. [...] The size of the LNPs was characterized as 44 ± 4 nm using Zetasizer Nano-ZS, and mRNA encapsulation efficiency was determined to be $93 \pm 2\%$ using the Quant-iT RiboGreen RNA assay and the Infinite M200 microplate reader (excitation/emission wavelengths:480/520 nm).”

Figure 1 (FOR REVIEW ONLY). Biodistribution of fluorescence (DiR)-labeled mRNA-SM-102 LNPs prepared *via* Moderna formulation (i.e., Moderna SM-102; molar ratio of SM-102:cholesterol:DSPC:DMG-PEG = 50:38.5:10:1.5) and modified formulation (i.e., Modified SM-102; molar ratio of SM-102:cholesterol:DSPC:DMG-PEG = 48:40:10:2) in different

organs. Both quantitative (a) and representative (b) images were obtained by IVIS 24 h post injection with an mRNA dose of 0.25mg kg^{-1} .

Figure 2 (FOR REVIEW ONLY). Comparison of mScarlet3 expression in different organs using mRNA-MPN NPs and mRNA-SM-102 LNPs prepared by the Moderna formulation and modified formulation. Quantitative data was obtained from IVIS. Five mice were included in each group and the quantitative data were normalized to DPBS-treated C57BL/6J mice. Statistical significance was analyzed using two-way ANOVA with Šídák’s multiple comparisons test: ns, $P > 0.05$, not significantly different.

Regarding the suggestion of using firefly luciferase-encoding mRNA, this mRNA is not suitable for our mRNA-MPN NP platform as EGCG may act as an inhibitor to luciferase. We have included the associated simulation results (Supplementary Fig. 19 in the original SI (Supplementary Fig. 20 in the revised SI)) and explanation as follows:

Page 8 of the original (page 9 of the revised) manuscript:

“To analyze FLuc transfection, immunofluorescence staining was conducted instead of a typical bioluminescence assay (i.e., based on the reaction between luciferase and luciferin substrate to produce a luminescent product) as EGCG interferes with the assay (Supplementary Fig. 19).”

The caption of Supplementary Fig. 19 of the original SI (Supplementary Fig. 20 in the revised SI):

“The superposition of luciferin and EGCG (docked at the same grid) revealed a significant spatial overlap at the active site. Further structural analysis suggested that EGCG were coordinated by T343 and G316 by hydrogen bonds and stabilized by possible π - π stacking between H245 and the phenylic group. EGCG was docked at the pocket with a minimal affinity of $-7.58\text{ kcal mol}^{-1}$. We therefore rationalize that free EGCG may act as an inhibitor to luciferase, possibly through direct or combined competition.”

Furthermore, to minimize the interference of tissue auto-fluorescence, the DPBS-treated mice were included in all *in vivo* experiments as an auto-tissue fluorescence control. Spectral

unmixing was conducted using DPBS-treated mice to unmix the auto-tissue fluorescence in the final result.

To clarify this point, we have now updated *In vivo transfection using lead MPN NPs and mScarlet3* section in METHODS on page 23 of the revised manuscript as follows:

“[...] Major organs, i.e., liver, kidney, lung, heart, spleen, and brain, were harvested post injection, followed by IVIS imaging (PerkinElmer, USA). Spectral unmixing was conducted using DPBS-treated mice to unmix the auto-tissue fluorescence. Five biologically independent mice were included in each group.”

3. Fig. 2e, in vitro mRNA release experiment, how is the structural integrity of the released mRNA?

Response: We thank the reviewer for raising this point. We did not analyze the structural integrity of the release mRNA *in vitro* as conditions would differ from intracellular release. The *in vitro* expression of mRNA is often used instead to evaluate the integrity of mRNA (*Pharm. Med.* **2022**, *36*, 11) as mRNA needs to remain intact upon entering the cytosol for successful expression (*Int. J. Pharm.* **2021**, *601*, 120586). In Fig. 3b–i of the original (and revised) manuscript, we demonstrated the successful *in vitro* expression of different types of mRNA, including mCherry, FLuc, and NGFR, with superior transfection efficiency over RNAiMax, implying a high degree of structural integrity of the released mRNA.

For clarity, we have updated the discussion on page 10 of the revised manuscript as follows:

“[...], with both higher transfection efficiency (by 2.5-fold) and average level of protein expression in each cell (i.e., MFI; by ~3-fold) than RNAiMax (Fig. 3g–i). Collectively, these results indicate some degree of structural integrity of the mRNA released from mRNA-MPN NPs and demonstrate the broad applicability of these NPs as an mRNA delivery platform.”

4. Fig. 6f-h, the authors highlight that the organ tropism of the NPs is influenced by the type of metal ions, can the authors comment on the potential mechanism? I wondered whether the incorporation of different metal ions may alter the physical-chemical properties of the NPs.

Response: We thank the reviewer for raising this point. We speculate that the metal ion-mediated organ deposition is likely driven by the biomolecular corona. Varying the type of metal ions leads to compositional changes on mRNA-MPN NPs, thereby altering the composition of the biomolecular corona formed on the NP surface (*Nano Lett.* **2020**, *20*, 2660). This variability in corona composition may influence the organ tropism of mRNA expression (*Proc. Natl. Acad. Sci. U.S.A.* **2021**, *118*, e2109256118).

We have now addressed this point on page 18 of the revised manuscript as follows:

“[...], and lung (i.e., ~11%, 4.5-fold higher radiance efficiency than DPBS-treated mice). Varying the type of metal ions leads to compositional changes on mRNA-MPN NPs, potentially altering the composition of the biomolecular corona formed on the NP surface as shown previously with MPN-based NPs⁵⁴. This variability in corona composition may influence the organ tropism of mRNA expression, which was also observed with LNPs⁵⁵.”

REVIEWER #3

In their manuscript, the Caruso et. al. report a versatile, non-cationic nanoparticle platform consisting of a poly(ethylene-glycol)-polyphenolic network that can package and deliver mRNA payloads. In brief, the authors develop a library of metal-organic nanoparticles with robust mRNA transfection in vitro and in mice (mRNA-MPN NPs). Key studies in the manuscript detail the assembly of their mRNA-MPN NP platform using formulation screening and optimization, characterization of the physiochemical properties of their mRNA-MPN NPs, in vitro evaluation of their mRNA-MPN NP platform, in vivo evaluation of their mRNA-MPN NP platform (including expression studies relative to Moderna's SM-102 LNP, biocompatibility studies, and further in vivo performance studies detailing mRNA expression using their highly performing formulations). Overall, the work by Caruso et. al. is innovative, interesting, and thorough. Their system is both well-characterized and affords interesting mRNA delivery properties that are distinct from the SM-102 formulations. Notably, their mRNA-MPN NP platform also delivers mRNA payloads to the brain following IV injections, opening novel therapeutic avenues for mRNA therapies. Further, the manuscript is well-written and well-organized, and it is the opinion of this reviewer that this manuscript will be of interest to the broader mRNA delivery community. Based on this, this reviewer recommends publication with minor revisions following consideration of the following points:

1) Page 5, Line 171 - the authors note that: "...increasing the concentration of metal ions beyond a mass ratio of 1:40 inversely impacted the transfection efficiency (73% and 4% at Zr^{IV}-to-EGCG mass ratios of 3:40 and 10:40, respectively) despite the higher mRNA loading observed at higher Zr^{IV} concentrations." Could the authors provide further commentary on this observation and speculate as to why this may be the case?

Response: We thank the reviewer for raising this point. We speculated on this observation on page 5 of the original manuscript as follows:

"[...] despite the higher mRNA loading observed at higher Zr^{IV} concentrations (Fig. 1h inset and Supplementary Fig. 3e, f). The observed reduction in MFI with higher amounts of Zr^{IV} within mRNA-MPN NPs (Fig. 1h) may be indicative of hindered mRNA release or fluorescence quenching (Supplementary Fig. 8)."

To demonstrate the hindering of mRNA release, mRNA-MPN NPs with different Zr^{IV}-to-EGCG mass ratios (i.e., 1:40, 2:40, 3:40, 10:40) were incubated in complete DMEM at 37 °C for 24 h. Consistent with the trend observed in our *in vitro* mRNA transfection, mRNA release was reduced at Zr^{IV} concentrations beyond mass ratios of 1:40 (e.g., ~0.75 µg mRNA was released at a mass ratio of 1:40 and the amount of released mRNA decreased to ~0.55 and ~0.3 µg at mass ratios of 2:40 and 10:40, respectively). We speculate that excess metal ions further increase the cross-linking density of mRNA-MPN NPs, impeding the cytosolic release of mRNA and subsequent transfection.

The results are shown as Supplementary Fig. 7b–d (new figure parts) in the revised SI, with accompanying discussion on page 5 of the revised manuscript as follows:

"We next investigated the effect of the metal ion-to-phenolic ligand (EGCG) mass ratio [...] mass ratio of 100. Compared to the NPs with no metal ion, the inclusion of Zr^{IV} with Zr^{IV}-to-EGCG mass ratios up to 1:40 gradually increased the transfection efficiency from 78 to 95%, and the MFI by 2.5-fold (Fig. 1g, h, and Supplementary Fig. 7). The integration of metal ions

also increased the mRNA loading (Fig. 1h inset and Supplementary Fig. 3e, f). However, increasing the concentration of metal ions beyond a mass ratio of 1:40 inversely impacted the transfection efficiency (reducing to 4% at a Zr^{IV}-to-EGCG mass ratio of 10:40) despite the higher mRNA loading observed at higher Zr^{IV} concentrations (Fig. 1h inset and Supplementary Fig. 3e, f). To investigate this further, mRNA release corresponding to different Zr^{IV}-to-EGCG mass ratios was evaluated using fluorescently labelled mRNA. As shown in Supplementary Fig. 7b–d, mRNA release decreased at Zr^{IV} concentrations beyond mass ratios of 1:40 (i.e., ~0.75 µg mRNA was released at 1:40, whereas only ~0.55 and ~0.3 µg mRNA was released at 2:40 and 10:40, respectively). This decrease is consistent with the reduction observed in *in vitro* mRNA transfection at the same ratios. This suggests that higher amounts of Zr^{IV} within mRNA-MPN NPs (i.e., Zr^{IV}-to-EGCG mass ratio >1:40) may increase the cross-linking density of mRNA with NPs, thereby hindering mRNA release and reducing the efficacy of mRNA transfection. In addition, a higher metal concentration is likely to induce fluorescence quenching (Supplementary Fig. 8), thus further reducing the MFI of transfected cells.

2) Figure 4c – it is a bit difficult for this reviewer to understand the key take away messages from this figure. Are there additional ways that this data may be represented (even if in the supporting information) that may improve comprehension for some readers?

Response: We thank the reviewer for pointing this out. We have added a supporting figure (Supplementary Fig. 27) in the revised SI to clarify the difference in mScarlet3 expression in all harvested organs between mRNA-MPN NPs and SM-102 LNPs (Moderna formulation).

The associated discussion was updated on page 12 of the revised manuscript as follows:

“[...], as well as overall higher expression across all harvested organs relative to the mRNA-SM-102 LNPs at the same mRNA dosage (Fig. 4c, d, and Supplementary Figs. 26 and 27).”

3) Figure 4d – the delivery profile for mRNA-MPN NPs to the brain is exciting. One question from this reviewer is about the background signal observed in the brain from DPBS. Could the authors provide additional brief commentary on where this background signal may arise from?

Response: We thank the reviewer for raising this point. The background signal in the brain of DPBS-treated mouse is likely tissue autofluorescence. The excitation and emission wavelengths of mScarlet3 are 561 and 594 nm, respectively. Within this range of wavelengths, tissue autofluorescence has been observed. However, as we show in Fig. 4d of the original and revised manuscript, despite the auto fluorescence, the fluorescence intensity of the brain from mRNA-MPN NP-treated mice is significantly higher (8-fold) than DPBS-treated mice, indicating expression in the brain of NP-treated mice. Therefore, the presence of tissue autofluorescence does not change our conclusions.

For clarity, we updated the caption of Fig. 4 in the revised manuscript as follows:

“Fig.4 [...] c,d, Comparison of mScarlet3 expression using mRNA-MPN NPs and mRNA-SM-102 LNPs: both quantitative [...] mean ± SD. The fluorescence signal observed in DPBS-treated mice may be due to tissue autofluorescence, which is often observed at the excitation and emission wavelengths of 561 and 594 nm, respectively. Statistical [...]”

4) Figure 4d – once again, the delivery profile of mRNA-MPN NPs to the brain is exciting. Could the authors provide further commentary speculating on how/why their mRNA-

MPN NPs are better able to deliver mRNA to the brain than the SM-102 formulation? Also, could the authors provide further commentary as to what potential therapeutic avenues these results may open?

Response: We thank the reviewer for the suggestion. The organ tropism of NPs, including brain deposition, may be primarily mediated by the NP composition.

Based on the compositional mass ratio of mRNA-MPN NPs (e.g., lead formulation PEG:mRNA:EGCG:Zr^{IV} = 100:1:100:2.5), EGCG is one of the most predominant constituents of the NPs. EGCG has been shown to transiently enhance the blood–brain barrier permeability (*J. Funct. Foods*, **2020**, *65*, 103732; *J. Controlled Release* **2019**, *301*, 62; *Biochem. Biophys. Rep.* **2017**, *9*, 180), hence providing a mechanism for enhanced brain localization and transfection.

We proposed this possible mechanism and included the associated references on page 11 of the original manuscript as follows:

“This observation could be attributed to the ability of EGCG to transiently enhance the blood–brain barrier permeability,^{45–47} facilitating the entry of NPs loaded with mRNA into the brain.”

SM-102 LNPs typically show predominant accumulation in the liver and spleen, and their compositions may not be conducive to significant brain deposition.

For clarity, we have now provided further commentary on the possible reason and updated the associated discussion on page 13 of the revised manuscript as follows:

“[...] beyond the blood–brain barrier. This brain deposition may primarily be mediated by the composition of the mRNA-MPN NPs. Based on the compositional mass ratio of the lead NP formulation (i.e., PEG:mRNA:EGCG:Zr^{IV} = 100:1:100:2.5), EGCG is one of the most predominant constituents in mRNA-MPN NPs. EGCG has been shown to transiently enhance the blood–brain barrier permeability^{49–51}, which may provide a possible mechanism for the entry of mRNA-MPN NPs into the brain.”

Regarding the comment on the potential therapeutic avenues these results may open, we now provide a brief perspective on page 13 of the revised manuscript as follows:

“These findings present exciting opportunities to deliver mRNA to the brain, which, to our knowledge, has remained elusive with current mRNA delivery systems. Although further work is required to identify the cell types transfected in the brain, brain deposition enabled by mRNA-MPN NPs opens up avenues for the development of therapeutics for the possible prevention and treatment of brain tumors and neurological disorders (e.g., Parkinson’s disease, Alzheimer’s disease). Moreover, studies have shown that delivering polyphenols to the brain can modulate microglia-mediated inflammation and exert neuroprotective effects^{52,53}, hence providing further motivation for the application of mRNA-MPN NPs in the brain.”